# Introducing Brønsted acid sites to accelerate the bridging-oxygen-assisted deprotonation in acidic water oxidation

Yunzhou Wen [1,6], Cheng Liu[2,6], Rui Huang [1], Hui Zhang [3], Xiaobao Li[3], F. Pelayo García de Arquer [4], Zhi Liu[3,5], Youyong Li [2] ✉ & Bo Zhang [1] ✉

Oxygen evolution reaction (OER) consists of four sequential proton-coupled electron transfer steps, which suffer from sluggish kinetics even on state-of-the-art ruthenium dioxide ($RuO_2$) catalysts. Understanding and controlling the proton transfer process could be an effective strategy to improve OER performances. Herein, we present a strategy to accelerate the deprotonation of OER intermediates by introducing strong Brønsted acid sites (e.g. tungsten oxides, $WO_x$) into the $RuO_2$. The Ru-W binary oxide is reported as a stable and active iridium-free acidic OER catalyst that exhibits a low overpotential (235 mV at 10 mA cm$^{-2}$) and low degradation rate (0.014 mV h$^{-1}$) over a 550-hour stability test. Electrochemical studies, in-situ near-ambient pressure X-ray photoelectron spectroscopy and density functional theory show that the W-O-Ru Brønsted acid sites are instrumental to facilitate proton transfer from the oxo-intermediate to the neighboring bridging oxygen sites, thus accelerating bridging-oxygen-assisted deprotonation OER steps in acidic electrolytes. The universality of the strategy is demonstrated for other Ru-M binary metal oxides (M = Cr, Mo, Nb, Ta, and Ti).

The oxygen evolution reaction (OER) is one of the pivotal reactions in electrochemical energy storage and conversion[1], which is the anodic reaction in water electrolysis[2], $CO_2$ electroreduction[3], metal-air batteries[4,5], electro-winning[6], etc. Specifically, the proton-exchange membrane (PEM) water electrolysis devices require OER catalysts with high activity and corrosion resistance in acidic environments[7]. However, the sluggish kinetics of OER leads to high overpotentials. Even for well-studied benchmark ruthenium oxide ($RuO_2$) catalysts[8], the long-term catalytic activity is far less than the targets required for large-scale renewable energy conversion devices[7].

The conventional OER mechanism on $RuO_2$ can be described as four sequential proton-coupled electron transfer (PCET) deprotonation

steps, in which the protons are desorbed from the oxo-intermediates (and water molecular) and released into the electrolyte directly[9]. In alkaline solutions, the abundant $OH^-$ ions assist this direct deprotonation process[10,11]. In acidic conditions, however, direct deprotonation becomes difficult due to the high proton concentration in the electrolyte. Accelerating the deprotonation of oxo-intermediates is one promising direction to improve OER kinetics in acidic electrolytes.

Recent research on $RuO_2$ and $IrO_2$ systems showed that the bridging oxygen (denoted as $O_{bri}$ in the following text, a schematic of different oxygen sites is shown in Supplementary Fig. 1) can accept protons from $H_2O$ or OER intermediates, providing a new possible path to OER intermediates deprotonation through the participation of

[1]State Key Laboratory of Molecular Engineering of Polymers, Department of Macromolecular Science, Fudan University, Shanghai 200438, China. [2]Institute of Functional Nano & Soft Materials (FUNSOM) and Jiangsu Key Laboratory for Carbon-Based Functional Materials & Devices, Soochow University, Suzhou 215123, China. [3]State Key Laboratory of Functional Materials for Informatics, Shanghai Institute of Microsystem and Information Technology, Chinese Academy of Sciences, Shanghai 200050, China. [4]ICFO - Institut de Ciències Fotòniques, The Barcelona Institute of Science and Technology, Barcelona 08860, Spain. [5]School of Physical Science and Technology and Center for Transformative Science, ShanghaiTech University, Shanghai 201210, China. [6]These authors contributed equally: Yunzhou Wen, Cheng Liu. ✉e-mail: yyli@suda.edu.cn; bozhang@fudan.edu.cn

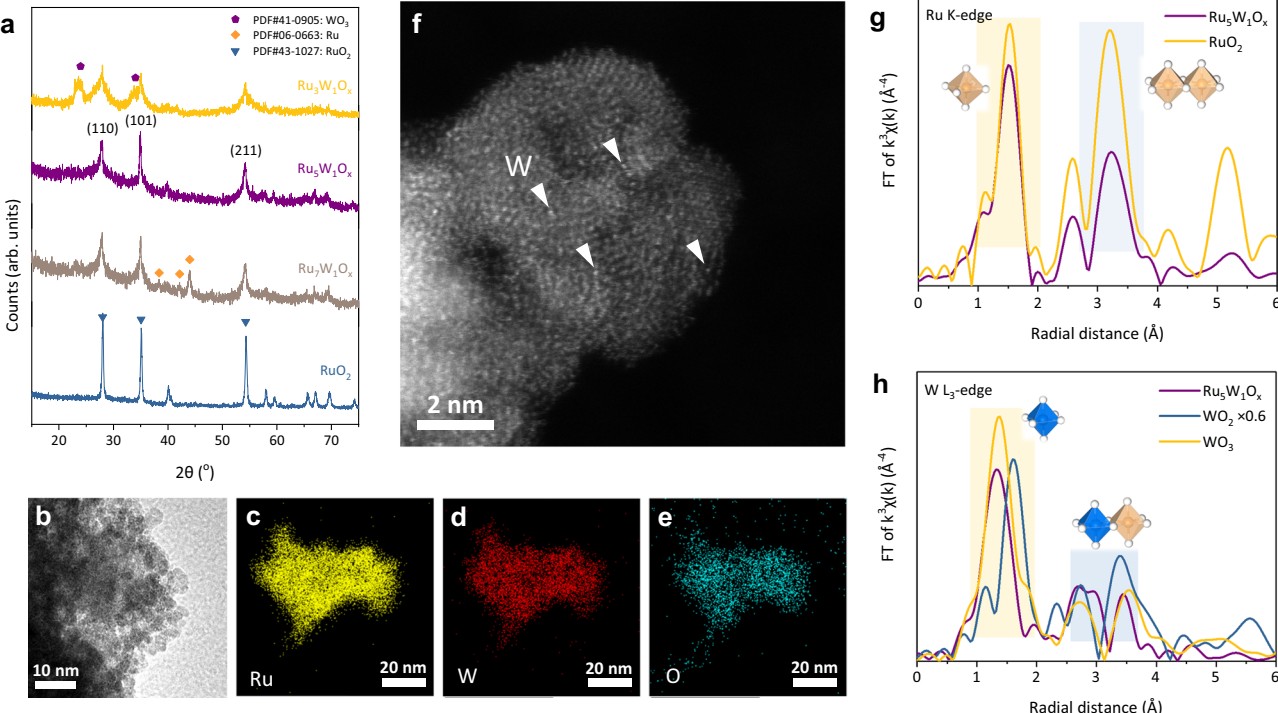

**Fig. 1 | Morphology of Ru-W oxides. a** The XRD patterns of different catalysts. No peaks from the segregated phases ($WO_3$ or metallic Ru) were observed in the pattern of $Ru_5W_1O_x$. **b** The HR-TEM image of as-prepared $Ru_5W_1O_x$ catalyst. **c–e** The EDX element mapping of Ru, W, and O. **f** The atomic resolution HAADF-STEM image of $Ru_5W_1O_x$. The bright spots were W atoms. **g**, **h** The $k^3$-weighted FT-EXAFS spectra of Ru K-edge and W $L_3$-edge. The $Ru_5W_1O_x$ had a shorter W-O distance than other W oxides, indicating a dense packing local structure. While tungsten oxides possess loose packing structures, this shortened W-O distance verified the incorporation of W into rutile $RuO_2$ lattice. The orange and blue octahedrons represent $RuO_6$ and $WO_6$ octahedrons, respectively.

$O_{bri}$[8,12,13]. A recent study on single-crystal $RuO_2$ has shown that on the $RuO_2$ (110) facet, the OOH* intermediate transfers one proton to the adjacent $O_{bri}$, forming protonated bridging oxygen ($OH_{bri}$) and the deprotonation of $OH_{bri}$ is the rate-determining step (RDS)[8,13]. By switching the facet orientation, the proton adsorption energetics on $O_{bri}$ can be tuned, thus altering the OER activity. However, the facet engineering approach is intrinsically limited to single crystals. Implementing this fundamental finding to improve the performance of industrially scalable and stable catalysts is still an open challenge[14]. Strategies to regulate the proton adsorption/desorption energetics on $O_{bri}$ and further accelerate this bridging-oxygen-assisted deprotonation (BOAD) process are urgently needed for the development of acidic OER electrocatalysts.

The deprotonation of surface $OH_{bri}$ sites can be described by the Brønsted-type acidity. In heterogeneous solid-acid catalysts such as zeolites[15], supported catalysts[16], and metal-organic frameworks[17], the acidity and density of Brønsted acid sites strongly affect the activity and mechanism of dehydration, isomerization, and cracking reactions. Similarly, it is rational that the deprotonation energetics of surface $O_{bri}$ sites can be optimized by precisely tuning the Brønsted acidity of $OH_{bri}$, thus the OER kinetics.

We, therefore, hypothesized that a tailored introduction of strong Brønsted acid sites into the $RuO_2$ lattice could optimize the deprotonation energetics of $O_{bri}$ on the catalytic surface. We implemented this strategy through the selective incorporation of tungsten (W) oxides –which have versatile crystal structure[18], acid stability[19], and unique proton adsorption[20,21]–to produce flexible surface $O_{bri}$ sites on $RuO_2$.

In this work, we successfully synthesize the Ru-W binary oxide catalysts achieving atomic-level uniform metal dispersion via the sol-gel method. The optimized catalyst demonstrates a 20-fold improvement of intrinsic OER activity compared to pristine $RuO_2$, which also achieves robust stability for more than 550 h of continuous electrolysis

with only 0.014 mV h$^{-1}$ degradation. Electrochemical studies, ex-situ/in-situ near-ambient pressure X-ray photoelectron spectroscopy (NAP-XPS) and density functional theory (DFT) calculations prove that the forming of W-$O_{bri}$-Ru Brønsted acid sites mitigates the too strong proton adsorption energy on $O_{bri}$ of $RuO_2$ and enables an easier proton transfer from the oxo-intermediates to the neighboring $O_{bri}$, and thus accelerated the overall acidic OER kinetics. Finally, the universality of such a strategy is confirmed in other Ru-M binary metal oxides (M = Cr, Mo, Nb, Ta, and Ti).

## Results

### Synthesis and characterizations of Ru-W binary oxide catalysts

We began with the synthesis of the Ru-W binary oxide catalysts via a modified sol-gel method (see Methods). By adjusting the feed ratio of metal precursors, we finally obtained the rutile $Ru_5W_1O_x$ catalyst with no obvious phase separation, as shown by X-ray diffraction patterns (XRD) (Fig. 1a). The high-resolution transmission electron microscopy (HR-TEM) showed that the as-prepared catalyst was 4–5 nm nanoparticles (Fig. 1b), with a Brunauer-Emmett-Teller (BET) surface area of 53.86 m$^2$ g$^{-1}$ (Supplementary Fig. 2). The element mapping of Energy-dispersive X-ray spectroscopy (EDX) confirmed the homogeneous distribution of Ru, W, and O in the materials (Fig. 1c–e). The spherical aberration corrected high-angle annular dark-field scanning transmission electron microscopy (HAADF-STEM) image (Fig. 1f) showed uniformly dispersed bright spots on the nanoparticle, which came from the atomic dispersion of W atoms into the $RuO_2$ lattice. The solid solution feature of $Ru_5W_1O_x$ was further confirmed by extended X-ray absorption fine structure (EXAFS). According to the Fourier transformed Ru K-edge EXAFS (FT-EXAFS) spectra, the rutile structure was maintained after W incorporation (Fig. 1g). The W atoms demonstrated a completely different coordination environment from common tungsten oxides, with a shorter W-O distance being observed than the

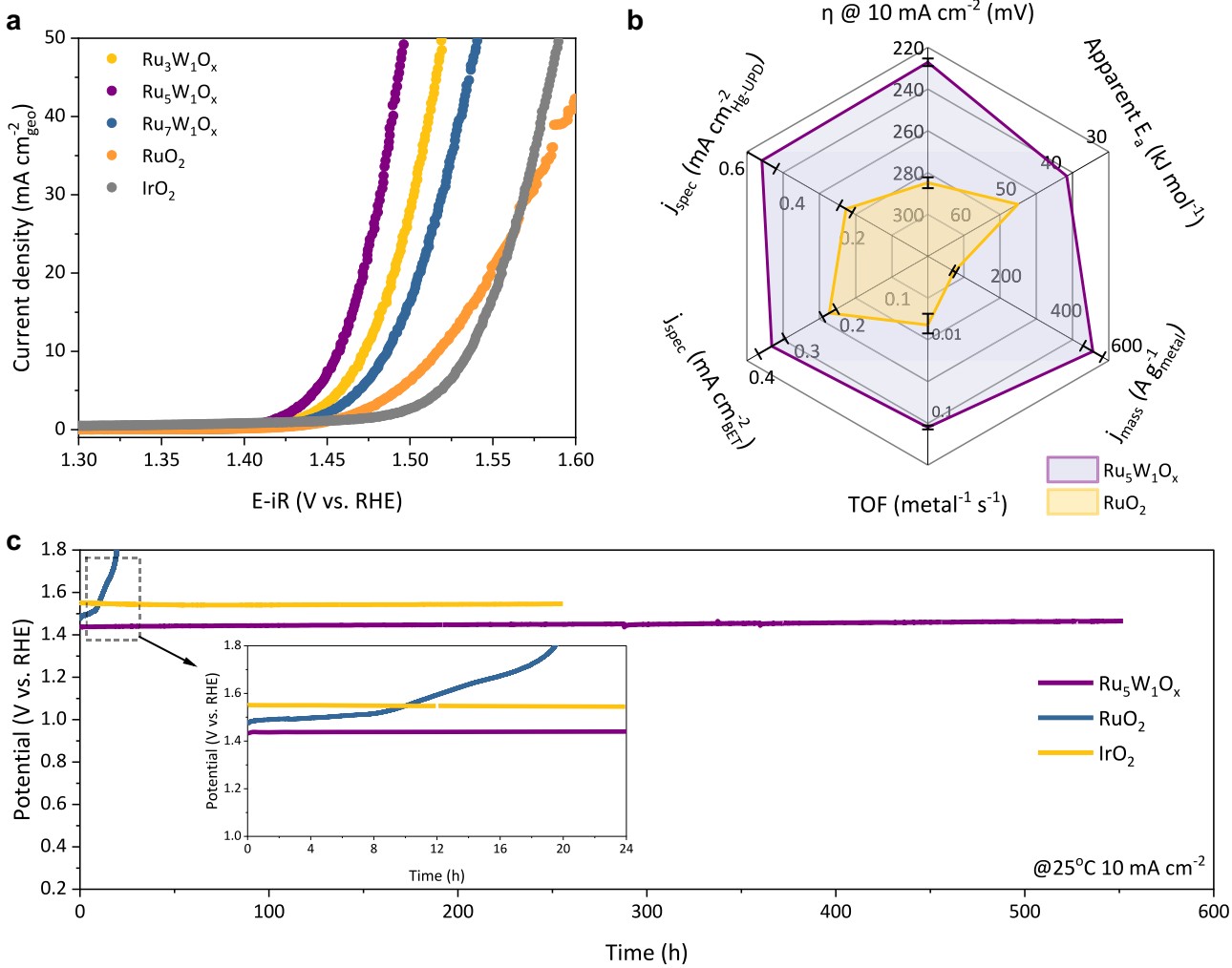

**Fig. 2 | Electrochemical performance of Ru-W oxides. a** The LSV curves of different catalysts with 95% iR compensation. Scan rate: 5 mV s$^{-1}$. **b** Summary of some major OER performance metrics of Ru$_5$W$_1$O$_x$ and RuO$_2$. The specific OER activity ($j_{spec}$) (normalized by BET surface area and Hg-UPD surface area respectively) was calculated at 1.50 V *vs.* RHE. The apparent activation energy ($E_a$) was calculated by the OER current of 1.50 V *vs.* RHE at different temperatures. The TOF and the mass-specific activity were calculated at $\eta = 300$ mV based on total metal loading. The error bars are standard deviations of averaging three independent measurements. **c** The stability comparison between Ru$_5$W$_1$O$_x$, RuO$_2$, and IrO$_2$. The stability of catalysts was evaluated by chronopotentiometry at 10 mA cm$^{-2}$.

WO$_3$ standard (Fig. 1f and Supplementary Fig. 3). The wavelet transformed EXAFS spectra of Ru$_5$W$_1$O$_x$ showed a distinct peak at **R** ≈ 3.5 Å, **k** ≈ 11 Å$^{-1}$, which could be attributed to the W-Ru scattering peak (Supplementary Fig. 4). The Raman spectroscopy of Ru$_5$W$_1$O$_x$ demonstrated a diminished rutile $B_{2g}$ mode (706 cm$^{-1}$) and a peak rising at 771 cm$^{-1}$, confirming the formation of W-O$_{bri}$-Ru structure in the catalyst (Supplementary Fig. 5). All the results above confirmed the atomically dispersed Ru-W solid solution oxide.

## Evaluation of OER performances

We then evaluated the OER performance of Ru$_5$W$_1$O$_x$ in a three-electrode system using 0.5 M H$_2$SO$_4$ as the electrolyte. All electrode potential was converted to the reversible hydrogen electrode (RHE). The linear sweep voltammetry (LSV) (Fig. 2a) of Ru$_5$W$_1$O$_x$ showed that the catalyst only needed 227 mV overpotential ($\eta$) to reach 10 mA cm$^{-2}$ current density −58 mV lower than the commercial nano-RuO$_2$ (Sigma-Aldrich, ~20 nm nanoparticles, Supplementary Fig. 6). To systematically compare the OER activity of different catalysts, several other performance metrics were also measured and calculated (Fig. 2b, Supplementary Table 1, and Supplementary Note 1). The OER performance Ru$_5$W$_1$O$_x$ outperformed the nano-RuO$_2$ at all six considered dimensions: The mass-specific activity was improved by 8-fold (750 A g$_{Ru}^{-1}$ of Ru$_5$W$_1$O$_x$ vs. 87 A g$_{Ru}^{-1}$ of RuO$_2$, estimated by total Ru loading

mass). When calculated by the total metal loading (Ru+W), the mass-specific activity of Ru$_5$W$_1$O$_x$ was 547 A g$_{metal}^{-1}$, 6 times higher than the RuO$_2$ (Supplementary Fig. 8). The turnover frequency (TOF) of Ru$_5$W$_1$O$_x$ reached 0.163 ± 0.010 s$^{-1}$ (at $\eta = 300$ mV), which was a 20-fold improvement from the pristine RuO$_2$ (0.007 ± 0.002 s$^{-1}$) (Supplementary Fig. 9). The specific activity of Ru$_5$W$_1$O$_x$ was obtained by normalizing the OER current using either the catalyst's BET surface area or the mercury underpotential deposition (Hg-UPD) determined electrochemical active surface area (ECSA) (Supplementary Fig. 10). Both values surpassed the pristine RuO$_2$ by *ca.* 2 times at 1.50 V vs. RHE (Supplementary Fig. 11). The apparent activation energy ($E_a$) was reduced from 42.2 kJ mol$^{-1}$ to 28.4 kJ mol$^{-1}$ after W incorporation (Supplementary Fig. 13). The above results verified that the incorporation of W-O$_{bri}$-Ru Brønsted acid sites improved the OER activity of RuO$_2$ both apparently (by the increase of electroactive surface area) and intrinsically (by the increase of activity of per active site), indicating a lower barrier and a different OER mechanism for Ru$_5$W$_1$O$_x$.

We next examined the OER stability of Ru$_5$W$_1$O$_x$ in acid using chronopotentiometry at 10 mA cm$^{-2}$. The catalyst showed no obvious activity decrease in the long-term operation (Fig. 2c). The overpotential was maintained at 235 mV after 550 h of continuous electrolysis, demonstrating a degradation rate of only 0.014 mV h$^{-1}$, which acted as a highly active iridium-free catalyst in long-term operation in

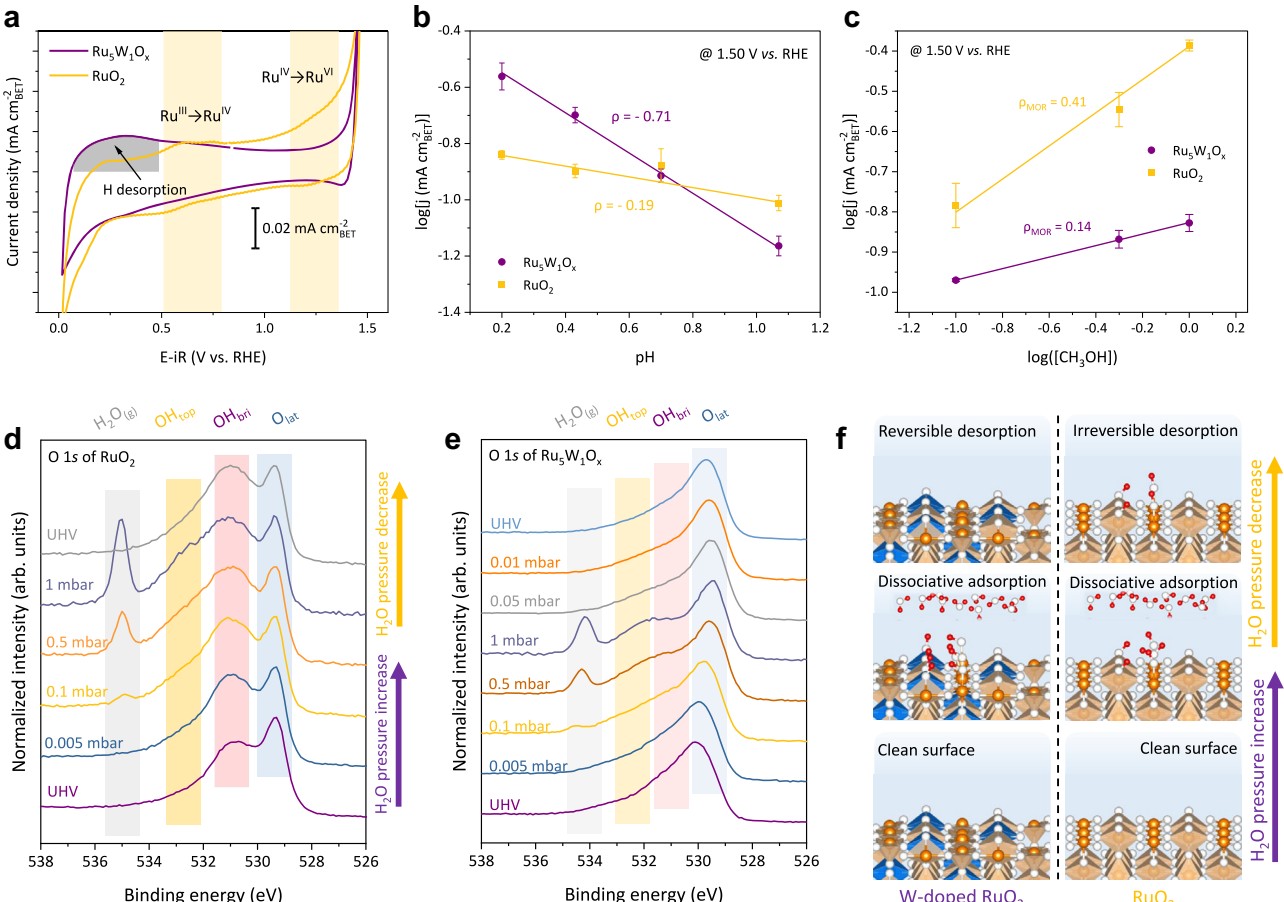

**Fig. 3 | Investigation of deprotonation on bridging oxygen. a** Typical CV curves of $Ru_5W_1O_x$ and $RuO_2$. Scan rate: 200 mV s⁻¹. **b** Logarithm of OER activity at 1.50 V vs. RHE as a function of pH. **c** The logarithm plots between MOR current density and concentration of methanol on different catalysts. The current densities in **a**–**c** were normalized by BET surface area. The error bars in **b**, **c** are standard deviations of averaging three independent measurements. **d, e** The O 1s XPS spectra of $RuO_2$ and $Ru_5W_1O_x$ at different water vapor pressures. Annotations: $O_{lat}$ - lattice oxygen, $OH_{bri}$ – protonated bridging oxygen, $OH_{top}$ – adsorbed or liquid water molecules, $H_2O_{(g)}$ – gas-phase water molecules. **f** A schematic demonstrated the surface species changes at different water vapor pressures. Orange balls – Ru, Blue balls – W, White balls – O, Red balls – H. The orange and blue octahedrons represent $RuO_6$ and $WO_6$ octahedrons, respectively.

acid electrolytes (Supplementary Fig. 14 and Supplementary Table 8). We also performed the same test on the commercial $IrO_2$ catalyst (~5 nm, BET surface area 11.98 m² g⁻¹, Supplementary Fig. 15), which was highly stable as expected. And the $Ru_5W_1O_x$ has shown comparable stability to state-of-the-art $IrO_2$ catalysts. In the intense cycle test, the $Ru_5W_1O_x$ could also stay active even after 20,000 CV cycles (Supplementary Fig. 16). While the $RuO_2$ showed poor stability in both chronopotentiometry and cycle tests (Supplementary Fig. 17). The morphology and composition of the $Ru_5W_1O_x$ catalyst didn't change significantly after electrolysis, as demonstrated by the HR-TEM images and EDX elemental mappings after OER (Supplementary Fig. 18). The in situ EXAFS also indicated that the W-O-Ru structure was retained under OER conditions (Supplementary Fig. 19). At higher electrolysis current densities (100 mA cm⁻²), the stability of $Ru_5W_1O_x$ was also maintained within a 100-h test (Supplementary Fig. 20). These data showed that the $Ru_5W_1O_x$ catalyst is a promising candidate for practical application.

### Investigation of deprotonation on bridging oxygen

To investigate the protonation/deprotonation on the catalyst's surface, we then conducted a series of electrochemical experiments correlated with proton transfer. We first examined the cyclic voltammetry (CV) profile differences between $Ru_5W_1O_x$ and $RuO_2$ (Fig. 3a). The CV of $RuO_2$ included two pairs of redox peaks at ca. 0.65 V and 1.25 V vs. RHE, which were often attributed to $Ru^{III}/Ru^{IV}$ and $Ru^{IV}/Ru^{VI}$

surface redox transitions, respectively[22]. In contrast, in $Ru_5W_1O_x$, the peak at ca. 1.25 V became less prominent, while a large plateau located between 0 V and 0.4 V vs. RHE arose instead. This plateau was similar to the hydrogen desorption peak on $WO_3$ or Pt electrodes[20,23,24] (Supplementary Fig. 21), indicating that deprotonation of $Ru_5W_1O_x$ surface required a much lower potential than $RuO_2$. We also checked the electrochemical behavior of $Ru_5W_1O_x$ in 1 M $HClO_4$ (same pH as 0.5 M $H_2SO_4$). No obvious electrolyte effect could be observed, which indicates that the adsorption of sulfate will not interfere with the surface chemistry of Ru, different from the Ir-based catalysts[25,26] (Supplementary Fig. 22).

We then examined the correlation between electrolyte pH and OER activity on different catalysts at the RHE scale[27] (Supplementary Fig. 23). As shown in Fig. 4b, $Ru_5W_1O_x$ demonstrated pH-dependent OER activity, with a reaction order (ρ) of −0.71. While for $RuO_2$, the ρ is only −0.19, demonstrating a weak pH-dependence of OER activity, coinciding with the previous report[22]. This pH-dependent activity difference could be elaborated by the acidity of bridging hydroxyl: the proton dissociation constant ($pK_a$) of Ru-$OH_{bri}$-Ru >> pH and the $O_{bri}$ sites of $RuO_2$ were saturated by protons within the experimental pH range. Whereas the W-$OH_{bri}$-Ru showed a stronger Brønsted acidity ($pK_a$ of $OH_{bri}$ close to pH), leading to a sensitive pH dependence of OER activity. Further verification of the Brønsted acidity of W-$OH_{bri}$-Ru sites was given by the ¹H solid-state nuclear magnetic resonant (¹H-NMR) spectrum, in which a split peak

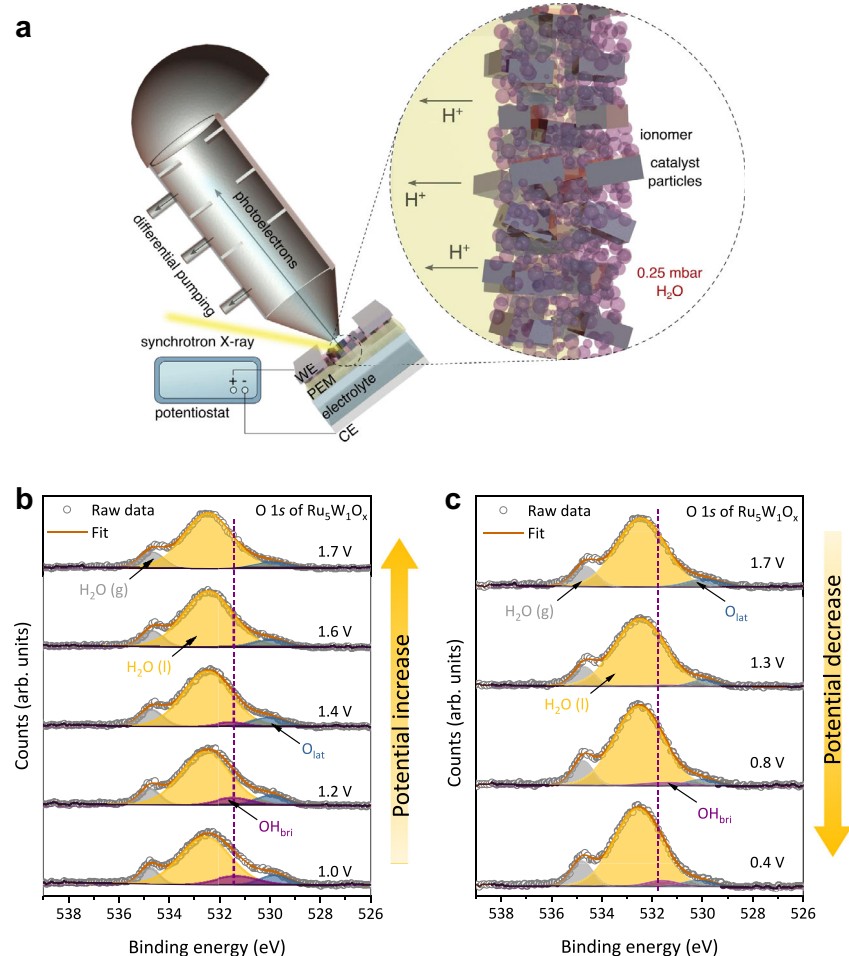

**Fig. 4 | In situ spectroscopic investigation of OER process. a** A schematic of the in situ electrochemical measurements. The zoom area illustrates the major components at the measured electrochemical interface. **b, c** In situ electrochemical O $1s$ XPS spectra of $Ru_5W_1O_x$. As the potential increased, the $OH_{bri}$ peak decreased accordingly, indicating the deprotonation of $W$-$OH_{bri}$-$Ru$ sites during OER. The $O_{bri}$ sites were re-protonated when decreasing the potential. The binding energy of all spectra was calibrated according to the Au $4f$ peak at 84.0 eV. The pressure of the XPS chamber was maintained at 0.25 mbar by injecting water vapor.

was observed indicating the formation of different $OH_{bri}$ sites[28,29] (Supplementary Fig. 24).

We next measured the surface OH* coverage on both catalysts using methanol as a molecular probe[30,31]. The methanol oxidation reaction (MOR) has a well-established mechanism that methanol molecules tend to nucleophilically attack the electrophilic OH*, so MOR will be more active on an OH* dominated surface[31]. We measured the reaction order of MOR on both catalysts (Fig. 3c and Supplementary Fig. 25) and found that $Ru_5W_1O_x$ is inert toward MOR, demonstrating a low surface OH* coverage. While $RuO_2$ showed higher MOR activity, indicating the $RuO_2$ surface was dominated by OH*. The above results verified that the deprotonation was easier on $Ru_5W_1O_x$ under applied potentials.

We finally analyzed the steady-state Tafel plots to study the apparent OER kinetics of different catalysts (Supplementary Fig. 26). The $RuO_2$ showed a 54 mV per decade (mV dec$^{-1}$) Tafel slope, suggesting that the reaction has a one-electron transfer electrochemical pre-equilibrium step (PES) followed by a pure chemical rate-determining step (RDS) with no electron transfer[7,21,22]. While $Ru_5W_1O_x$ showed a 42 mV dec$^{-1}$ slope. It corresponds to a one-electron transfer electrochemical PES followed by another one-electron transfer electrochemical RDS. We attributed these differences to the different proton binding energy on $O_{bri}$. Considering the BOAD pathway, protonated $Ru$-$OH_{bri}$-$Ru$ might inhibit the chemical proton transfer from the OER intermediates (OH* or OOH*) to the $O_{bri}$, while the less protonated $W$-$O_{bri}$-$Ru$ sites could favor the proton transfer to the $O_{bri}$, thus

shifting the RDS. The detailed deduction and discussion of the Tafel slope refer to Supplementary Note 2.

## Surface oxygen species study by NAP-XPS

To obtain further insights into the deprotonation process on $O_{bri}$, we then carried out NAP-XPS measurements under varied water vapor pressure (Supplementary Note 3). As shown in Fig. 3d, e, four different oxygen species were distinguished by O $1s$ XPS spectra: the lattice oxygen ($O_{lat}$) at *ca.* 530 eV, the protonated bridging oxygen ($OH_{bri}$) at *ca.* 531 eV, molecularly adsorbed water/hydroxyl ($OH_{top}$) adsorbed on the coordinatively unsaturated Ru sites ($Ru_{CUS}$) at *ca.* 533 eV and the gas phase water molecules ($H_2O_{(g)}$) at 534-535 eV[32]. Under ultra-high vacuum (UHV), $RuO_2$ showed a more than 3 times higher ratio of $OH_{bri}$ species than $Ru_5W_1O_x$ (Supplementary Fig. 29, Supplementary Table 3 and 4). With the increase in water vapor pressure, the ratio of $OH_{bri}$ in $RuO_2$ increased accordingly (Fig. 3d), which was contributed by the dissociative adsorption of water molecules on the $RuO_2$ surface[32] (Supplementary Fig. 30). However, when reduced the pressure to UHV, the $OH_{bri}$ ratio did not decrease accordingly, which verified the strong proton adsorption nature of $O_{bri}$ in the $Ru$-$O_{bri}$-$Ru$ structure.

In contrast, several different features were observed in the O $1s$ XPS spectra of $Ru_5W_1O_x$ (Fig. 3e). Firstly, the $O_{lat}$ peak positively shifted by ~0.5 eV, again proving the formation of Ru-W oxide solid solution[33]. The most prominent difference between $Ru_5W_1O_x$ and $RuO_2$ lies in the $OH_{bri}$ transition. The $OH_{bri}$ intensity did not change significantly along with the vapor pressure change. Instead, the $O_{lat}$ shifted to lower

binding energy, accompanied by valence changing of W from $W^{6+}$ to $W^{5+}$, as observed in W $4f$ XPS spectra (Supplementary Fig. 31 and Supplementary Table 5). These peak shifts were reversible when sequentially reducing the pressure back to UHV (Fig. 3e). The above results displayed a scenario that the deprotonation of water molecules (or oxo-intermediates during OER) was faster and more reversible in $Ru_5W_1O_x$ than that in $RuO_2$ (Fig. 3f). Detailed discussions refer to Supplementary Note 3.

The deprotonation process on $Ru_5W_1O_x$ was further monitored using an in-situ electrochemical NAP-XPS setup (Fig. 4a and Supplementary Fig. 32). As the potential increased, the $OH_{bri}$ peak decreased accordingly, demonstrating a potential-dependent deprotonation scenario (Fig. 4b). When reducing the applied potential, the $O_{bri}$ protonated and formed $OH_{bri}$ again (Fig. 4c), providing evidence of the reversible protonation/deprotonation nature of W-$O_{bri}$-Ru sites. On the contrary, due to the strong interaction between water and Ru-$O_{bri}$-Ru, the $RuO_2$ surface was covered by condensed water or $OH_{top}$ species and the deprotonation of $OH_{bri}$ could hardly be observed upon applying electrode potentials (Supplementary Fig. 35). The deprotonation process can also be verified by the in situ Raman spectroscopy (Supplementary Fig. 38). In $Ru_5W_1O_x$, the peak at $ca.$ 880 cm$^{-1}$ decreased along with the potential increase, indicating the deprotonation of W-$OH_{bri}$-Ru along with the potential increase. Detailed discussions on the in situ electrochemical XPS experiment refer to Supplementary Note 4.

### Theoretical investigation of the deprotonation energetics on Brønsted acid sites

To further understand the relationship between the Brønsted acidity of $O_{bri}$ and OER activity. We investigated the effect of introducing Brønsted acid sites into $RuO_2$ using DFT calculations. We inserted the $WO_6$ octahedrons into the stable $RuO_2$ (110) facet and constructed two types of $O_{bri}$ sites: Ru-$O_{bri}$-Ru and W-$O_{bri}$-Ru[34] (Fig. 5a inset). We then examined the adsorption energy ($E_{ads}$) of hydrogen atoms (by assuming the energy of $H^+ + e^-$ as the energy of 1/2 $H_2$ molecule) on different $O_{bri}$ sites. The Ru-$O_{bri}$-Ru demonstrated strong adsorption energy of H with an $E_{ads}$ of −1.04 eV, while the W-$O_{bri}$-Ru showed mild H adsorption energy ranging from −0.39 eV to −0.50 eV (Fig. 5a). This indicated that protons tended to spontaneously adsorb onto Ru-$O_{bri}$-Ru sites in acidic electrolytes. Thus extra energy input was needed to deprotonate the proton-saturated Ru-$OH_{bri}$-Ru sites under OER conditions for pristine $RuO_2$. In contrast, the low H adsorption energy on W-$O_{bri}$-Ru enables much easier deprotonation of the $OH_{bri}$ (a stronger Brønsted acidity). Since the deprotonation of $OH_{bri}$ was regarded as the rate-limiting step in Ru-based catalysts at low overpotential[8,35], we further calculated the kinetic barrier of the deprotonation on different $O_{bri}$ sites considering the effect of solvent (Supplementary Note 5, Supplementary Fig. 43 and 44). The W-$OH_{bri}$-Ru model showed a lower barrier of deprotonation compared with Ru-$OH_{bri}$-Ru, indicating a faster deprotonation process on W-$OH_{bri}$-Ru (Fig. 5b). All these DFT results coincided with the electrochemical and XPS measurements, which well explained how the Brønsted acid sites promoted the OER kinetics. To further understand the pH-dependent activity of the Ru-W catalyst, we checked the $E_{ads}$ of protons on W-$O_{bri}$-Ru with all Ru-$O_{bri}$-Ru saturated by protons (Supplementary Fig. 40). The $E_{ads}$ of protons kept reducing along with the increase of hydrogen coverage and finally reached nearly thermal-neutral adsorption energy (−0.06 eV), indicating high proton mobility of W-doped $RuO_2$ in strong acidic electrolytes.

By integrating the above electrochemical, spectroscopic, and theoretical results, we finally proposed an OER pathway including BOAD steps on the W-doped $RuO_2$ catalyst (Fig. 5c). In this mechanism, the $O_{bri}$ played a critical role in water dissociation and oxo-intermediates deprotonation. At each step, the adsorbed oxo-intermediate (or water molecule) first chemically transfers a proton

to the neighboring W-$O_{bri}$-Ru site, afterwards, the $OH_{bri}$ deprotonates accompanying an electron transfer. We calculated the thermodynamic OER overpotential of $Ru_5W_1O_x$ based on both the BOAD pathway and conventional adsorbates evolution mechanism (AEM) pathway using DFT[9]. The BOAD pathway showed an overpotential of 0.41 V while the AEM showed an overpotential of 0.78 V–a 0.37 V improvement by the BOAD mechanism.

### The universality of BOAD steps

To extend our strategy of regulating Brønsted acidity of $O_{bri}$ in acidic water oxidation, we further replaced W with other metals (M = Cr, Mo, Nb, Ta, and Ti), which are often used as Brønsted acids or bases, to form rutile-type oxides and examined their OER performances (Fig. 6a). The hydrogen adsorption energy of M-$O_{bri}$-Ru sites was also calculated using DFT. We found a linear relationship between the OER activity (represented by the TOF of Ru atoms) and the $E_{ads}$ of H atoms on $O_{bri}$ sites (Fig. 6b). The results showed that increasing the acidity of the $O_{bri}$ site on $RuO_2$ could lead to easier deprotonation and accelerate the BOAD process, which confirms the validity of our modulating strategy.

In summary, in this work, we demonstrated a strategy to modify the Brønsted acidity of bridging oxygen sites in $RuO_2$ to improve acidic water oxidation. The incorporation of Brønsted acid sites (e.g. $WO_x$) could optimize the proton adsorption energy of bridging oxygen sites. The electrochemical, in-situ and ex-situ X-ray spectroscopic and theoretical studies proved that: these W-$O_{bri}$-Ru bridging oxygen sites increased the mobility of protons on the catalyst surface and led to a fast bridging-oxygen-assisted deprotonation process, thus accelerating the OER kinetics. This strategy was proved to be universal in other Ru-M binary metal oxides (M = Cr, Mo, Nb, Ta, and Ti), and all catalysts demonstrated an excellent linear relationship between the OER activity and the $E_{ads}$ of protons on $O_{bri}$ sites. This work provides new insights into the OER mechanism and broadens the designing principles for new high-performance electrocatalysts.

## Methods
### Computational methods
All the calculations were performed by periodic DFT with the Vienna Ab-initio Simulation Package (VASP) code[36]. The projector augmented wave (PAW) method was used to describe the interaction between the atomic cores and electrons[37,38]. The kinetic energy cut-off for the plane-wave expansion was set to 400 eV, and the Brillouin-zone integrations for the adsorption model were sampled using a $(3 \times 3 \times 1)$ Monkhorst–Pack mesh[39]. The generalized gradient approximation (GGA) with PBE functional was used[40]. The convergence thresholds of the energy change and the maximum force for the geometry optimizations were set to $10^{-6}$ eV and 0.03 eV/Å, respectively. A vacuum of 15 Å in the z-direction was employed to avoid the interactions between periodic images.

For the chemisorption on W modified $RuO_2$ (110) surface, many possible adsorption configurations were explored, and the thermodynamic stability of different structures was determined by the adsorption energy ($\Delta E_{ads}$) that was defined as,

$$\triangle E_{ads} = E_{*M} - E_* - E_M \tag{1}$$

where $E_{*M}$ and $E_*$ represent the total energies of catalyst surface with and without adsorbate, respectively; $E_M$ is the total energy of adsorbate. All of them are available from the DFT calculation.

The computational hydrogen electrode (CHE) model[41] was employed to calculate the Gibbs free energy change ($\Delta G$) for each elementary reaction step and construct the free energy diagram for the

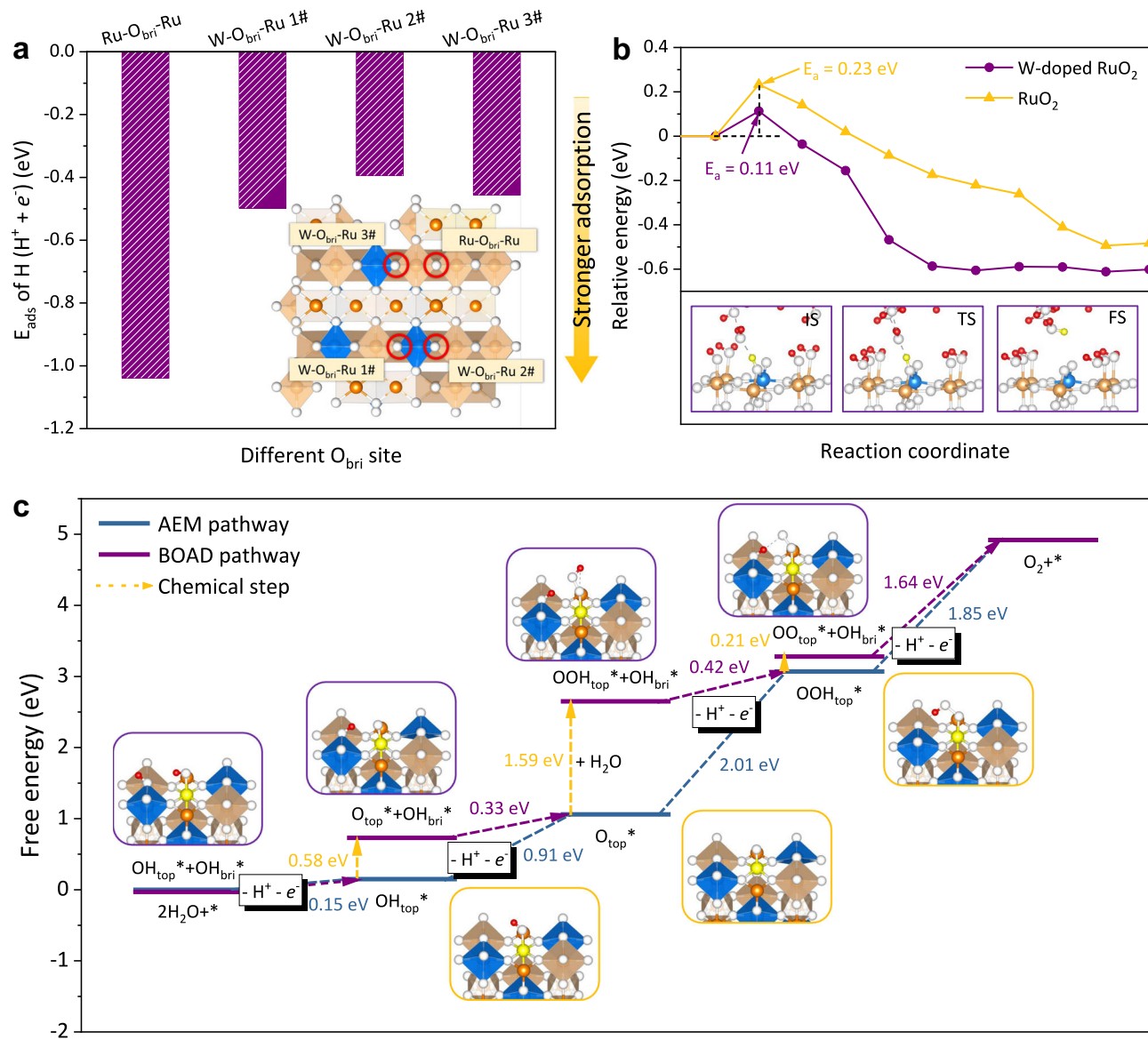

**Fig. 5 | DFT investigation of hydrogen adsorption on Ru-W binary oxides. a** The H atom adsorption energy on different surface $O_{bri}$ sites. Inset: Schematic of different $O_{bri}$ sites on the W-doped $RuO_2$. **b** The kinetic barrier of the deprotonation of $OH_{bri}$ on different catalysts with solvent. Insets: The snapshots of the initial state (IS), transition state (TS), and final state (FS) on W-doped $RuO_2$. **c** The free energy diagram of W-$RuO_2$ with different OER pathways. The active Ru site is marked yellow. Orange balls – Ru, Blue balls – W, White balls – O, Red balls – H. The orange and blue octahedrons represent $RuO_6$ and $WO_6$ octahedrons, respectively.

OER. The $\Delta G$ was computed using the following:

$$\triangle G = \triangle E + \triangle ZPE - T\triangle S + \triangle G_U + \triangle G_{pH} \qquad (2)$$

where $\Delta E$ is the reaction energy between the initial state and the final state of the elementary reaction, which is available from DFT total energy; the correction of zero-point energy ($\Delta ZPE$) and entropy at $T = 298.15\,K$ ($T\Delta S$) can be obtained from vibrational frequency calculations. $\Delta G_U = nU$, where $n$ and $U$ stand for the number of electrons transferred and the applied electrode potential, respectively. $\Delta G_{pH} = k_B T \times ln10 \times pH$, where $k_B$ is the Boltzmann constant. The free energy change between $1/2\,H_2$ and $H^+ + e^-$ will be zero at the potential of $0\,V$ and $1/2\,G_{(H2)}$ can be equal to the free energy of proton and electron.

To simulate the interaction at the water/(W-doped)$RuO_2$ interface, we used 18 explicit water molecules (6 layers) on a $2 \times 1$ $RuO_2$ surface slab (3 layers) with an area of $6.28 \times 6.42\,\text{Å}^2$. The simulation box

is $28\,\text{Å}$ along the z-axis. The initial structure of the water box is based on the density of the solvent[42,43] (as shown in Supplementary Fig. 43). To equilibrate the waters interacting with the interface, we carried out 850 steps (time step is 1 fs) of ab initio molecular dynamics (AIMD) simulation at 298 K[44]. The temperature and potential energies during the AIMD simulation are shown in Supplementary Fig. 44. To calculate deprotonation barriers of adsorbed H, we made use of the climbing image nudged elastic band (CI-NEB) method[45] based on the established models.

## Synthesis of catalysts
The Ru-W binary oxide catalysts were synthesized by a sol-gel method. In a typical procedure, 0.75 mmol ruthenium trichloride hydrate ($RuCl_3 \cdot xH_2O$, Sigma-Aldrich) and 0.15 mmol tungsten hexachloride ($WCl_6$, Sigma-Aldrich) were first dissolved in 3 mL N, N-dimethylformamide (DMF) and cooled in a refrigerator for 2 h. Then 200 μL of deionized water was added. In the meantime, 500 μL

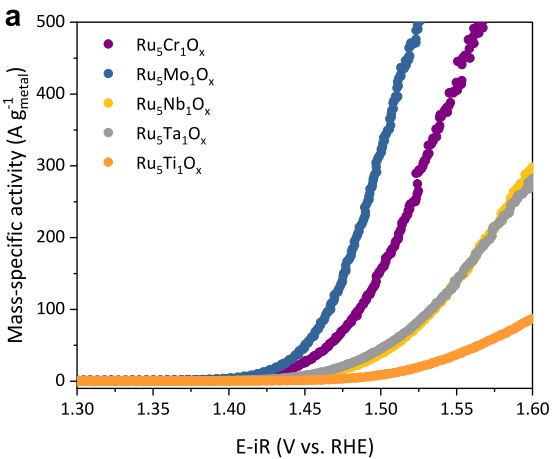

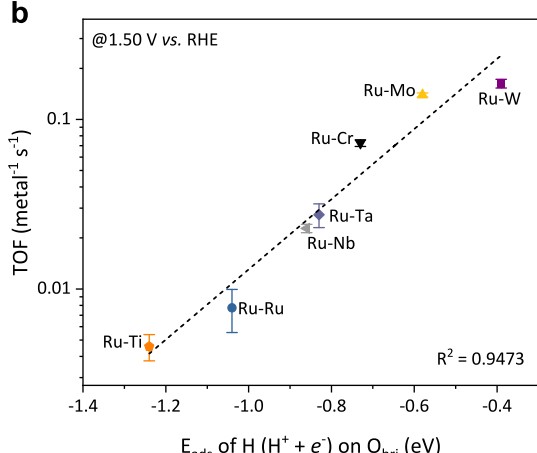

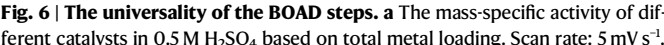

**Fig. 6 | The universality of the BOAD steps. a** The mass-specific activity of different catalysts in 0.5 M $H_2SO_4$ based on total metal loading. Scan rate: 5 mV s$^{-1}$. **b** The TOF value of different catalysts (regarding all metal atoms were active sites) as a function of H adsorption energy on $O_{bri}$ sites.

propylene oxide (Sigma-Aldrich) was dropwise added into the solution using a syringe pump under stirring. The solution was then placed and aged overnight before the reaction was quenched by adding acetone. The formed precipitations were washed with acetone 3 times and dried in the vacuum. The dried powder was then grounded and annealed at 500 °C for 1 h to obtain the final catalyst.

To synthesize other reference catalysts, the same procedure was used by adjusting the ratio of precursors (the total amount of metal precursors was controlled at 0.9 mmol) or changing the metal precursors. The synthesis of Ru-M (M = Cr, Mo, Nb, Ta, and Ti) followed the same procedure with $Ru_5W_1O_x$. Chromium chloride hexahydrate ($CrCl_3·6H_2O$), molybdenum chloride ($MoCl_5$), niobium chloride ($NbCl_5$), tantalic chloride ($TaCl_5$), and titanium tetrachloride ($TiCl_4$) (all purchased from Sigma-Aldrich) were used as metal precursors. The $RuO_2$ nanoparticles (~25 nm) were purchased from Sigma-Aldrich without further treatment. The commercial $IrO_2$ nanoparticles (~20 nm) were purchased from PERIC Inc.

## Materials characterizations

The X-ray diffraction (XRD) patterns of prepared catalysts were measured by a Bruker D8A diffractometer. The Brunauer-Emmett-Teller surface area of the catalysts was obtained by a Quantachrome Autosorb-iQ analyzer. An FEI Tecnai G20 transmission electron microscope (TEM) was used to obtain the high-resolution TEM (HR-TEM) images and corresponding energy-dispersive X-ray spectroscopy (EDX) elemental mapping were from with an Oxford energy disperse spectrometer. The spherical aberration corrected high-angle annular dark-field scanning transmission electron microscopy (HAADF-STEM) images were obtained by a Titan G2 300 kV TEM (Thermo Fisher Scientific).

The hard (Ru K-edge and W $L_3$-edge) X-ray Absorption spectroscopies (XAS) measurements were carried out at the 1W1B beamline of the Beijing Synchrotron Radiation Facility (BSRF), respectively. The XAS data were processed, normalized, and fitted using the Demeter software package embedded with FEFF 8.5 codes[46]. The wavelet transformation of EXAFS spectra was performed by WTEXAFS software[47].

The $^1$H solid-state nuclear magnetic resonance ($^1$H-NMR) spectroscopy was performed on a Bruker 400WB AVANCE III spectrometer. The catalysts powder was dehydrated at 300 °C in the air for 2 h before measurements.

## Electrochemistry

The evaluation of electrochemical performance was carried out in a three-electrode system. All electrolytes were purged by argon to

remove the dissolved oxygen during measurements. To prepare the working electrode (WE), 5 mg catalyst powder, 2 mg carbon black (XC-72), 980 μL mixed solution (water: ethanol = 5:1, v/v), and 20 μL Nafion solution (5 wt%, Sigma-Aldrich) were mixed and sonicated to form a homogeneous ink. Then, 4.5 μL ink (catalyst loading = 0.0225 mg) was drop-cast onto a clean glassy carbon rotating disk electrode (RDE, Autolab, 3 mm in diameter) and dried at room condition. All electrochemical measurements were carried out by a Metrohm Autolab PGSTAT204 potentiostat. A saturated mercurous sulfate electrode (MSE, $E_O$ = 0.652 V vs. RHE) was used as the reference electrode (RE) and a Pt foil was used as the counter electrode (CE). The measured potential was calibrated to the RHE scale according to:

$$E_{RHE} = E_{Hg_2SO_4} + 0.652 + 0.0591 \times pH \tag{3}$$

To evaluate the OER activity of different catalysts, the WEs were first performed 10 CV cycles between 0.95 to 1.50 V vs. RHE (before iR-correction) at a 50 mV s$^{-1}$ scan rate to clean and stabilize the surface, then followed with an LSV scan from 0.95 to 1.65 V at 5 mV s$^{-1}$ scan rate and 2500 rpm rotation speed. For the pH-dependent activity measurement, 0.05, 0.1, 0.25 and 0.5 M $H_2SO_4$ solution (pH = 1.1, 0.7, 0.4, 0.2, respectively. Measured by a Horiba D-71 pH meter) was used as electrolyte without adding buffer salt. The methanol oxidation was measured in 0.5 M $H_2SO_4$ containing different concentrations of methanol. The steady-state Tafel slope was measured by elevating the potential from 1.25 to 1.75 V vs. RHE by 20 mV each step. Each step was maintained for 10 s. The uncompensated solution resistances ($R_\Omega$) were measured by extrapolating the electrochemical impedance semi-circle to the high-frequency end, which was ca. 7 Ω for each electrode in 0.5 M $H_2SO_4$.

The turnover frequency values were calculated according to the following equation:

$$TOF = \frac{j \times A \times \eta}{4 \times e \times n} \tag{4}$$

where $j$ is the current density at 1.53 V vs. RHE after 95% iR compensation, $A$ is the geometric area of GCE (0.0706 cm$^2$), $\eta$ is the Faradic efficiency and $e$ is the charge of an electron (1.602 × 10$^{-19}$ C) and $n$ is the number of active sites.

The active site number $n$ was determined by assuming all Ru atoms (or all metal atoms) as active sites (underestimating case),

according to the following equation:

$$n_{mass} = \frac{m_{loading} \times N_A}{Mw} \times n_{metal} \qquad (5)$$

where $m_{loading}$ is the loading mass of the catalyst. $N_A$ is Avogadro's constant ($6.022 \times 10^{23}$ mol$^{-1}$), $Mw$ is the molecular weight of catalysts (estimated by the molecular formula $Ru_5W_1O_{13}$ for $Ru_5W_1O_x$) and $n_{metal}$ is the number of Ru atoms or metal atoms per molar of catalysts.

The stability measurements were carried out by air-brush spraying the catalysts powder onto the carbon paper (TGP-H-060, Toray). The catalyst loading was *ca.* 1.5 mg cm$^{-2}$. The electrolysis cell was kept in a 25 °C constant temperature water bath. 100 μL of water was added to the cell every four days to keep the concentration of the electrolyte constant.

### Evaluation of the electrochemical active surface area (ECSA)

In this paper, we used mercury underpotential deposition[48,49] (Hg-UPD) and electrochemical double-layer capacitance ($C_{dl}$) to evaluate the ECSA of different catalysts. To prepare the WE, 2 mg catalysts powder and 1 mg XC-72 carbon black were sonicated in 2 mL water/ethanol mix solution containing 20 μL Nafion solution and 3 μL obtained ink was drop-casted on the RDE. The catalyst loading was 42.5 μg cm$^{-2}$.

For the Hg-UPD method, the fresh electrode was first cycled in Ar-purged 0.1 M HClO$_4$ at −0.15 to +0.65 V vs. MSE to obtain the background (50 mV s$^{-1}$, 1600 rpm). Then, the same electrode was moved to an Ar-purged 0.1 M HClO$_4$ electrolyte containing 1 mM Hg(NO$_3$)$_2$ (Alfa Aesar) and cycled under the same condition. The current difference of the cathodic scans between the Hg-containing solution and blank background was integrated to calculate the amount of Hg$_{upd}$. A coulombic charge of 138.6 μC cm$^{-2}$ was used as a factor to obtain the Hg$_{upd}$-ECSA values.

For the double-layer capacitance method, the $C_{dl}$ values were obtained by conducting CV cycles at various scan rates from 20 mV s$^{-1}$ up to 200 mV s$^{-1}$ in Ar-purged 0.5 M H$_2$SO$_4$. The CVs were scanned between 0.20 and 0.30 V vs. MSE. The cathodic and anodic charging currents measured at 0.25 V vs. MSE (close the open circuit potential) were plotted as a function of scan rate. The average slope of the anodic and cathodic plot is the $C_{dl}$ value. A general specific capacitance ($C_s$) of 35 μF cm$^{-2}$ was used to calculate the $C_{dl}$-derived ECSA[50].

### Near-ambient pressure X-ray photoelectron spectroscopy (NAP-XPS)

The AP-XPS spectra were measured at the BL02B01 beamline of Shanghai Synchrotron Radiation Facility[51] (SSRF). The incident photon energy was set to 735 eV to distinguish the surface species. To measure the adsorption isotherm of water vapor, the powder catalysts were first tableted and mounted into the analysis chamber. Before the measurements, the catalysts were first heated to 250 °C under 0.1 mbar O$_2$ atmosphere for 30 min to remove the adsorbed water and carbon species. Then, the chamber was pumped back to UHV and returned to room temperature. Ru 3$d$, O 1$s$, and W 4$f$ XPS spectra were collected at this stage and regarded as the initial state. In the following experiments, different amount of water vapor was injected into the chamber successively, and the XPS spectra were measured under different conditions. For each catalyst, two independent measurements were performed to ensure the validity of the results. Other details of the NAP-XPS experiment are described in Supplementary Note 2.

### In situ electrochemical XPS

The in situ electrochemical XPS experiments were also carried out at the BL02B01 of SSRF, using a homemade electrochemical cell. The design of the electrochemical cell was similar to the cell reported by Falling et al.[52]. The cell was equipped with a gold-coated titanium lid as the WE and a Pt foil as the CE and RE. A Nafion 117 proton-exchange membrane (PEM) was used to seal the electrolyte from the vacuum. To prepare the sample, the interested catalysts were first spray-coated onto the PEM and hot-pressed at 140 °C, then boiled the catalyst-coated membrane in 0.5 M H$_2$SO$_4$ and deionized water to remove the impurities. During the measurements, the cell was filled with 0.05 M H$_2$SO$_4$ as the cathodic electrolyte (the anodic electrolyte was the PEM). The pressure of the XPS chamber was maintained at 0.25 mbar by injecting water vapor to relieve the evaporation of electrolytes and provide reactant. The incident photon energy was set to 735 eV to distinguish the surface species. A Biologic SP-200 potentiostat was used to apply potentials. The CE of the cell was grounded to the electron energy analyzer so that the potential of the WE can be directly controlled by the potentiostat. At each potential, an Au 4$f$ spectrum on the lid was measured to calibrate the binding energy. Other details of the in situ XPS experiment are described in Supplementary Note 3.

### In situ Raman spectroscopy

The Raman spectra of the powder catalysts were measured either on a Horiba XploRA or a Renishaw In Via Qontor Raman spectrometer. The in situ electrochemical Raman spectroscopy was performed on a Horiba XploRA Raman spectrometer equipped with a 60× waterproof objective and a 638 nm laser. In the in situ measurements, a home-made electrochemical cell, equipped with a saturated Ag/AgCl reference electrode and a Pt wire counter electrode, was used. The spectra were collected at the steady-state under different applied potentials. Each spectrum was integrated for 10 s and averaged by two exposures.

### Reporting summary

Further information on research design is available in the Nature Research Reporting Summary linked to this article.

## Data availability

The authors declare that all data supporting the results of this study are available within the paper and its supplementary information files or from the corresponding author upon reasonable request. The electrochemical data of OER performances is provided as the Source Data in this paper. Source data are provided with this paper.

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

## Acknowledgements

This work was supported by NSFC (21875042, 21902179, and 22173067), STCSM (21DZ1207102 and 21DZ1207103), and the National Key R&D Program of China (Grant No. 2017YFA0204800). This work was also supported by the Program for Eastern Scholars at Shanghai Institutions. The authors thank BL02B01 of SSRF supported by NSFC No. 11227902. The authors thank Prof. Liqiang Zhang of Yanshan University for the spherical aberration corrected HAADF-STEM experiment. H.Z. acknowledges the support of the Shanghai Sailing Program (Grant No. 19YF1455600). F.P.G.d.A. thanks the CEX2019-000910-S [MCIN/AEI/ 10.13039/501100011033], Fundació Cellex, Fundació Mir-Puig, the Generalitat de Catalunya through CERCA, and La Caixa Foundation.

## Author contributions

B.Z. and Y.W. conceived and designed the experiments. Y.W. and R.H. synthesized the materials and performed the electrochemical measurements. C.L.and Y.L. performed the DFT calculations and analysis. Y.W., X.B.L., H.Z., and Z.L. designed and participated in the in situ XPS measurements. F.P.G.d.A. participated in the discussion and interpretation of experimental and theoretical data. B.Z. and Y.W. wrote the manuscript. All of the authors discussed the results and commented on the manuscript.

## Competing interests

The authors declare no competing interests.
