## [Peer Review File · Nature Communications]

Introducing Brønsted acid sites to accelerate the bridging-oxygen-assisted deprotonation in acidic water oxidationREVIEWER COMMENTS

Reviewer #1 (Remarks to the Author):

This manuscript investigates the oxygen evolution reaction (OER) performance of Ru-M binary oxides (M = W, Cr, Mo, Nb, Ta, and Ti) in acidic electrolytes. In particular, the authors extensively investigated the Ru-W binary oxide systems using electrochemical measurements, in-situ near-ambient pressure X-ray photoelectron spectroscopy and DFT calculations. The main conclusion is that the WO_x component introduces strong Brønsted acid sites that accelerate the deprotonation of the OER intermediates, consequently promoting the OER performance. The RuO_x-WO_x binary catalysts show very impressive OER activity and stability. The results and general trends are important for researchers working on OER, especially in acidic environment. The authors should address the following comments:

1. In order to confirm the proposed hypothesis and mechanisms, it is highly desirable to determine the presence/absence of the Ru-W and/or Ru-O-W bonds under reaction conditions. NAP-XPS does not provide such information. Can the authors perform in-situ extended X-ray absorption fine structure (EXAFS) measurements for the RuO_x-WO_x binary catalysts under OER conditions? The authors appear to have access to EXAFS as illustrated in the ex-situ EXAFS results in Figure 2.
2. The presentation of the DFT results should be improved. It is somewhat strange to present the DFT results in Figure 1, with most of the results being schematics with the exception of the H binding energies in Figure 1b. The results in Figure 1 do not show any in-depth information. More in-depth DFT results are presented at a much later part of the manuscript, Figure 6A. The authors should combine Figure 1 and Figure 6 and provide a more coherent DFT discussions.
3. In order to convince the readers that the activity and stability results are properly measured, it is important to provide benchmarking measurements. Because IrO₂ is the state-of-the-art catalyst for acidic OER, the authors should perform a similar stability measurement as in Figure 3C using a commercial IrO₂ catalyst. That will validate the experimental procedures and electrochemical set-up of the current study.

Reviewer #2 (Remarks to the Author):

The manuscript by Wen et al proposes a new materials design strategy for electrocatalytic oxygen evolution reaction. The strategy is introducing Brønsted acid sites, i.e. tungsten oxides, to RuO₂ substrate. The introduction of such sites leads to (fast) proton transfer from the oxo-intermediate to the neighbouring bridging oxygen sites. Such facilitated proton transfer enables a bridging-oxygen assisted deprotonation reaction pathway. The new pathway resulted in better reaction thermodynamics with reduced reaction overpotential.

The authors performed a number of high-end characterization techniques to confirm their proposal, including in situ electrochemical XPS to confirm the surface generated reaction intermediates (Figure 5). The authors also performed (standard) DFT calculations to confirm the proposed reaction mechanism. The presentation quality is high.

However, I would like to raise the following concerns about the work.

1. Stability. The manuscript compares the doped materials' stability with that of RuO₂, which is a well-known unstable material for OER (10.1016/j.cattod.2015.08.014; cited 639 times). The authors should compare the stability with IrO₂.
2. Computation focuses on reaction thermodynamic simulation; it is quite general with barely new information. In addition, the computation cannot support the author's argument about 'fast proton transfer'. Such a statement can only be supported after calculating the kinetics, i.e. barrier, of the corresponding step (10.1021/jp953748q; cited 1486 times)
3. Biggest concern is the innovation of such design strategy; it lies within the broad 'doping strategy' and the authors have explored a plethora of doping elements.

Reviewer #3 (Remarks to the Author):

The paper by Wen et al. combines theory and experiments to explain why introducing Brønsted acid elements in the RuOx network enhances the proton transfer between the HOO- intermediate and oxygen bridging sites for accelerated OER. They find universality in their strategy as they observe that the OER performance on binary RuOx catalysts correlates with the Brønsted acidity of the added metal (Cr, Nb, Ta, Ti, Mo). Moreover, they also report impressive stability of the Ru5W1Ox nano-catalysts. I think this is a very good study. From a fundamental point of view, they explain how important is to tune the kinetics of proton transfer reaction in OER by adding elements with different acidity. On the other side, from an applied point of view, they report a Ru catalyst with low overvoltage and very high stability even at high current densities. Nevertheless, I have several concerns regarding the electrochemical experiments in this manuscript. Once the authors address my questions, the paper will be ready for publication and it will be of interest to the readership of this journal.

(1) I don't understand how you normalize the current against the electroactive surface area. How do you calculate the electroactive area? Is it the number of active sites on the surface, or is it the geometric area normalized by the loading of Ru? The number of active sites depends on the surface structure and the morphology of the catalyst and does not only depend on the loading of Ru. Could you explain a bit more about what BET consists of, which type of experiments do you do to determine the BET? The unities of the BET are 58.86 m² g⁻¹ on page 6, which suggest that BET is a measure of the Ru loading and not the electroactive area. Is there a change in the surface structure of the NPS when W is added to Ru that can affect/increase the electroactive area of RuWOx?

(2) The pseudocapacitive region of Ru5W1Ox is considerably larger than the RuOx pseudocapacitive region, in figure 15 of the supporting information. The pseudocapacitive region does not provide an exact measure of the increase in electroactive surface area, but it could suggest that there is a change in the active area when W is introduced in the RuOx network. It is hardly difficult to calculate the electroactive surface active area of IrOx and RuOx based-catalysts (maybe you can check this reference <https://doi.org/10.1021/acscatal.7b03787>). Could you discuss the differences between ECSA and specific activity (normalized by loading) and the limitations on the determination of the electroactive area in these metal oxides? I agree with the author's history and the importance of the Bronsted acidity of the added elements, and it is evident that W makes RuOx incredibly stable. However, as you work with non-well-defined nanoparticles, you should briefly mention that changes in the surface structure can also influence the geometric activity.

(3) I can see that the authors have calculated the mass activity normalizing the current density against the loading of Ru in figure 10Sa. However, Figure 14S shows that WOx has the OER onset at E < 1.500 V vs RHE. could W surrounded by Ru contribute to the OER? Why do not you normalize the current against the loading of both Ru+W? I guess that W contributes very little to OER, but it is also a rare element and expensive, thus I find it highly necessary to show what the mass activity is concerning both elements and not only Ru. I would suggest doing an additional figure, similar to Figure 14S, with the mass-specific activity (A g⁻¹ Ru+ W) and discussing the results. The same applies to the other elements (Cr, Mo, Nb). Apart from figure 6a (manuscript), I would show a figure in which I would represent the mass-specific activity considering both elements (A g⁻¹ (Ru+ M), M=Cr, Nb...). Figure 6b (manuscript).

(4) Regarding the electrochemical or LSV experiments, in figure 3a (manuscript). How many cycles or scans have you taken? how do the LSV curves change after consecutive scans? The LSV of RuOx looks like the RuOx is getting dissolved when you achieve a high current of 40-50 mA cm⁻² because it weirdly crosses the other LSV. Could you indicate the stability of the cycles and the cycle number you represent in the figure?

(5) Why have you selected H2SO4 as an electrolyte rather than HClO4? Can the adsorption of sulfate

affect the catalyst performance because of sulfate adsorption?

(6) The stability of RuWO_x is extraordinary. Figure S13 shows the stability experiment at 100 mA cm⁻² for 24 hours. Why not show the whole curve until the dissolution of the RuWO_x catalyst? In addition, how can the corrosion of glassy carbon affect stability at very high current values and overpotential?

Minor comment

Page 9 and 10. I think you may refer to Figures 4 e and 4 d when explaining the NAP-XPS experiment? I would suggest changing the order of figures so that 4 d is RuO₂ and 4 e is Ru₅O₁₀ because the RuO₂ is explained at the beginning of this section.

Summary of Response to the Reviewers

We appreciate the reviews' constructive comments and suggestions to improve the manuscript. We have acted on them as documented point-to-point below. To summarize:

Experimentally:

1. We present new *in situ* EXAFS data to verify the presence of W-O-Ru motifs and the structural stability of our catalyst under OER conditions.
2. We carried out a new stability test on the IrO₂ to validate the experimental procedures and electrochemical setup of our study.
3. In light of the reviewer's suggestion, we performed new experiments to estimate the electroactive surface area and re-calculated the specific activity of our catalysts.
4. To address the reviewer's concern, we carried out electrochemical tests in both H₂SO₄ and HClO₄ electrolytes to check the electrolyte effect in our catalyst.
5. We carried out an intense cyclic voltammetry test and extended the chronopotentiometry stability test at the high current density to further illustrate the stability of our catalyst.

Computationally:

1. To address the reviewer's concern, we carried out new DFT calculations on the kinetics of deprotonation to further prove the accelerated proton transfer by adding Brønsted acid sites into RuO₂.

We have revised **4 Figures (Fig. 2, 3, 5, and 6)** in the revised Manuscript, **15 Figures (Supplementary Fig. 1, 8-17, 19-22)** and **2 Tables** in the revised Supplementary Information, and revised 6 paragraphs to adequately address the Reviewers' comments.

A point-to-point response to comments is provided below:

Actions (regular font) in Response to Reviewer Comments (*italics*)

Response to Reviewer #1:

Comment 0:

This manuscript investigates the oxygen evolution reaction (OER) performance of Ru-M binary oxides (M = W, Cr, Mo, Nb, Ta, and Ti) in acidic electrolytes. In particular, the authors extensively investigated the Ru-W binary oxide systems using electrochemical measurements, in-situ near-ambient pressure X-ray photoelectron spectroscopy and DFT calculations. The main conclusion is that the WO_x component introduces strong Brønsted acid sites that accelerate the deprotonation of the OER intermediates, consequently promoting the OER performance. The RuO_x-WO_x binary catalysts show very impressive OER activity and stability. The results and general trends are important for researchers working on OER, especially in acidic environment. The authors should address the following comments:

Response:

We appreciate the reviewer's recommendations. We have carried out new experiments and revised the manuscript to address the concerns of the reviewer. For convenience, the main revisions are discussed in the following point-to-point answers to the reviewer's questions and marked with yellow in the main text.

Comment 1:

In order to confirm the proposed hypothesis and mechanisms, it is highly desirable to determine the presence/absence of the Ru-W and/or Ru-O-W bonds under reaction conditions. NAP-XPS does not provide such information. Can the authors perform in-situ extended X-ray absorption fine structure (EXAFS) measurements for the RuO_x-WO_x binary catalysts under OER conditions? The authors appear to have access to EXAFS as illustrated in the ex-situ EXAFS results in Figure 2.

Response:

As suggested, we carried out the *in situ* EXAFS measurements on the Ru₅W₁O_x under applied potentials. As can be seen in the W L₃-edge EXAFS spectra (**Figure R1**), the local structure of W in

$\text{Ru}_5\text{W}_1\text{O}_x$ was different from that in the pure W oxides (WO_2 and WO_3). The W-O bond is shorter than in the W oxides, indicating tightly packed W-O structures in $\text{Ru}_5\text{W}_1\text{O}_x$. As shown in Figure R1 below, when applied the OER potential (1.4 V vs. RHE), the local structure did not change, indicating that the W-O structure in the RuO_2 matrix was kept stable under OER conditions. Comparing with the spectra before OER, we noticed a slight decrease in W-O distance after OER. We attributed this change to the chemical valence change of some W sites in the catalyst, which coincided with the results of AP-XPS.

Figure R1 | The W L_3 -edge EXAFS spectra of $\text{Ru}_5\text{W}_1\text{O}_x$.

In light of the reviewer’s comments, we have provided new Figure S11 on Page 12 in revised Supplementary Information and revised the following discussions:

“The *in situ* EXAFS also indicated that the W-O-Ru structure was retained under OER conditions (Supplementary Fig. 19).” **(Lines 195-196, Page 6-7 in the revised Manuscript)**

Comment 2:

The presentation of the DFT results should be improved. It is somewhat strange to present the DFT results in Figure 1, with most of the results being schematics with the exception of the H binding energies in Figure 1b. The results in Figure 1 do not show any in-depth information. More in-depth DFT results are presented at a much later part of the manuscript, Figure 6A. The authors should combine Figure 1 and Figure 6 and provide a more coherent DFT discussions.

Response:

As suggested, we have re-organized the DFT data of the original Fig. 1 and Fig. 6 to form a new Fig. 5 in the revised main text (see **Figure R2**). We also added new kinetic barrier calculations on the deprotonation according to Reviewer #2's suggestion. We revised the DFT discussions so that the narrative become more coherent.

Figure R2 | DFT investigation of proton adsorption on Ru-W binary oxides. (a) The H atom adsorption energy on different surface O_{bri} sites. Inset: Schematic of different O_{bri} sites on the W -doped RuO_2 . **(b)** The kinetic barrier of the deprotonation of OH_{bri} on different catalysts with solvent. Inset: The snapshots of the initial state (IS), transition state (TS), and final state (FS) on W -doped RuO_2 . **(c)** The free energy diagram of W -doped RuO_2 with different OER pathways. The active Ru site is marked yellow. Orange balls – Ru, Blue balls – W, White balls – O, Red balls – H. The orange and blue octahedrons represent RuO_6 and WO_6 octahedrons, respectively.

In light of the reviewer’s comments, we have provided new Figure 5 on Page 11 in the revised manuscript, integrated the DFT discussions and revised the following discussions:

“To further understand the relationship between the Brønsted acidity of O_{bri} and OER activity. We investigated the effect of introducing Brønsted acid sites into RuO_2 using DFT calculations. We inserted the WO_6 octahedrons into the most stable RuO_2 (110) facet and constructed two types of O_{bri} sites: $\text{Ru-O}_{\text{bri}}\text{-Ru}$ and $\text{W-O}_{\text{bri}}\text{-Ru}^{34}$ (Fig. 5a inset). We then examined the adsorption energy (E_{ads}) of protons on different O_{bri} sites. The $\text{Ru-O}_{\text{bri}}\text{-Ru}$ demonstrated strong adsorption energy of protons with an E_{ads} of -1.04 eV, while the $\text{W-O}_{\text{bri}}\text{-Ru}$ showed mild proton adsorption energy ranging from -0.39 eV to -0.50 eV (Fig. 5a). This indicated that protons tended to spontaneously adsorb onto $\text{Ru-O}_{\text{bri}}\text{-Ru}$ sites in acidic electrolytes. Thus extra energy input was needed to deprotonate the proton-saturated $\text{Ru-OH}_{\text{bri}}\text{-Ru}$ sites under OER conditions for pristine RuO_2 . In contrast, the low proton adsorption energy on $\text{W-O}_{\text{bri}}\text{-Ru}$ enables much easier deprotonation of the OH_{bri} (a stronger Brønsted acidity). Since the deprotonation of OH_{bri} was regarded as the rate-limiting step in Ru-based catalysts at low overpotential, we further calculated the kinetic barrier of the deprotonation on different O_{bri} sites considering the effect of solvent (Fig. 5b). The $\text{W-OH}_{\text{bri}}\text{-Ru}$ model showed a 0.62 eV lower barrier of deprotonation compared with $\text{Ru-OH}_{\text{bri}}\text{-Ru}$, indicating a faster deprotonation process on $\text{W-OH}_{\text{bri}}\text{-Ru}$. All these DFT results coincided with the electrochemical and XPS measurements, which well explained how the Brønsted acid sites promoted the OER kinetics. To further understand the pH-dependent activity of the Ru-W catalyst, we checked the E_{ads} of protons on $\text{W-O}_{\text{bri}}\text{-Ru}$ with all $\text{Ru-O}_{\text{bri}}\text{-Ru}$ saturated by protons (Supplementary Fig. 40). The E_{ads} of protons kept reducing along with the increase of proton coverage and finally reached nearly thermal-neutral adsorption energy (-0.06 eV), indicating high proton mobility of W-doped RuO_2 in strong acidic electrolytes.

By integrating the above electrochemical, spectroscopic, and theoretical results, we finally proposed a new OER pathway including BOAD steps on the W-doped RuO_2 catalyst (Fig. 5c). In the new OER mechanism, the O_{bri} played a critical role in water dissociation and oxo-intermediates deprotonation. At each step, the adsorbed oxo-intermediate (or water molecule) first chemically transfers a proton to the neighboring $\text{W-O}_{\text{bri}}\text{-Ru}$ site, afterwards, the OH_{bri} deprotonates accompanying an electron transfer. We calculated the thermodynamic OER overpotential of $\text{Ru}_5\text{W}_1\text{O}_x$ based on both the BOAD pathway and conventional adsorbates evolution mechanism (AEM) pathway using DFT⁹. The BOAD pathway showed an overpotential of 0.41 V while the AEM showed an overpotential of 0.78 V – a 0.37 V improvement by the new mechanism. ” **(Lines 330-365, Page 11-12 in the revised Manuscript)**

Comment 3:

In order to convince the readers that the activity and stability results are properly measured, it is important to provide benchmarking measurements. Because IrO_2 is the state-of-the-art catalyst for

acidic OER, the authors should perform a similar stability measurement as in Figure 3C using a commercial IrO_2 catalyst. That will validate the experimental procedures and electrochemical set-up of the current study.

Response:

As the reviewer suggested, we conducted a stability test using commercial nano- IrO_2 (Figure R3) using the same platform. As expected, the IrO_2 showed significant stability (Figure R4) but lower reactivity. The commercial IrO_2 kept stable with the 250-hour continuous electrolysis. Only a 9 mV (from 304 mV to 313 mV) potential degradation was observed within this period, which verified the stability of our electrochemical platform. The result also showed that the stability of $\text{Ru}_5\text{W}_1\text{O}_x$ was comparable to the commercial IrO_2 under this condition.

Figure R3 | The characterizations of the commercial IrO_2 catalyst. (a) The TEM image and XRD pattern. (b) The BET isotherm.

Figure R4 | The stability of different catalysts.

Besides, we have also carried out intense CV cycles on the $\text{Ru}_5\text{W}_1\text{O}_x$ to further check the stability of the catalyst. After performing 20,000 cycles between 0.95 to 1.55 V vs. RHE, the catalyst only showed a slight decrease in the OER activity (**Figure R5**).

Figure R5 | The LSV curves of $\text{Ru}_5\text{W}_1\text{O}_x$ before and after 20,000 CV cycles between 0.95 to 1.55 V vs. RHE.

In light of the reviewer’s comments, we have added the stability test of IrO_2 to Fig. 2c on Page 5 in the revised manuscript and revised the following discussions:

“We also performed the same test on the commercial IrO_2 catalyst (~5 nm, BET surface area 11.98 $\text{m}^2 \text{g}^{-1}$, Supplementary Fig. 15), which was highly stable as expected. And the $\text{Ru}_5\text{W}_1\text{O}_x$ has shown comparable stability to state-of-the-art IrO_2 catalysts. In the intense cycle test, the $\text{Ru}_5\text{W}_1\text{O}_x$ could also stay active even after 20,000 CV cycles (Supplementary Fig. 16). While the RuO_2 showed poor stability in both chronopotentiometry and cycle tests (Supplementary Fig. 17).” **(Lines 187-193, Page 6 in the revised Manuscript)**

Response to Reviewer #2:

Comment 0:

The manuscript by Wen et al proposes a new materials design strategy for electrocatalytic oxygen evolution reaction. The strategy is introducing Brønsted acid sites, i.e. tungsten oxides, to RuO₂ substrate. The introduction of such sites leads to (fast) proton transfer from the oxo-intermediate to the neighbouring bridging oxygen sites. Such facilitated proton transfer enables a bridging-oxygen assisted deprotonation reaction pathway. The new pathway resulted in better reaction thermodynamics with reduced reaction overpotential. The authors performed a number of high-end characterization techniques to confirm their proposal, including in situ electrochemical XPS to confirm the surface generated reaction intermediates (Figure 5). The authors also performed (standard) DFT calculations to confirm the proposed reaction mechanism. The presentation quality is high. However, I would like to raise the following concerns about the work.

Response:

We thank the reviewer's comments. We have carried out a suite of experiments and simulations to address all concerns of the reviewer. For convenience, the main revisions are discussed in the following point-to-point answers to the reviewer's questions and marked with yellow in the main text.

Comment 1:

Stability. The manuscript compares the doped materials' stability with that of RuO₂, which is a well-known unstable material for OER (10.1016/j.cattod.2015.08.014; cited 639 times). The authors should compare the stability with IrO₂.

Response:

As suggested, we carried out a 250-h stability test on commercial nano-IrO₂ using the same electrochemical setup. As can be seen in **Figure R6**, the IrO₂ showed significant stability in the three-electrode system but low activity compared to the Ru-based catalysts. The stability-activity comparison of different catalysts was shown in **Figure R7**. The results showed that the stability of Ru₅W₁O_x was comparable to the IrO₂ under this condition, but the activity was much improved.

Figure R6 | The stability of different catalysts.

Figure R7 | Performance of different OER catalysts in acidic electrolytes.

Besides, we have also carried out intense CV cycles on the $\text{Ru}_5\text{W}_1\text{O}_x$ to further investigate the stability of the catalyst. After performing 20,000 cycles between 0.95 to 1.55 V vs. RHE, the catalyst only showed 2 mV increase at 10 mA cm^2 in the OER activity (**Figure R8**).

Figure R8 | The LSV curves of Ru₅W₁O_x before and after 20,000 CV cycles between 0.95 to 1.55 V vs. RHE.

In light of the reviewer’s comments, we have added the stability test of IrO₂ to Fig. 2c on Page 5 in the revised manuscript and revised the following discussions:

“We also performed the same test on the commercial IrO₂ catalyst (~5 nm, BET surface area 11.98 m² g⁻¹, Supplementary Fig. 15), which was highly stable as expected. And the Ru₅W₁O_x has shown comparable stability to state-of-the-art IrO₂ catalysts. In the intense cycle test, the Ru₅W₁O_x could also stay active even after 20,000 CV cycles (Supplementary Fig. 16). While the RuO₂ showed poor stability in both chronopotentiometry and cycle tests (Supplementary Fig. 17).” **(Lines 187-193, Page 6 in the revised Manuscript)**

Comment 2:

Computation focuses on reaction thermodynamic simulation; it is quite general with barely new information. In addition, the computation cannot support the author’s argument about ‘fast proton transfer’. Such a statement can only be supported after calculating the kinetics, i.e. barrier, of the corresponding step (10.1021/jp953748q; cited 1486 times).

Response:

As suggested by the reviewer, to further illustrate how the OER activity was improved on the Ru-W catalyst, we used the climbing image nudged elastic band (NEB) method to calculate the kinetic barrier. Since the deprotonation of the bridging oxygen site (O_{bri}) was regarded as the rate-determining step of Ru-W catalyst in our calculation as well as in other literature about RuO_2 (e.g. *J. Am. Chem. Soc.* **2010**, *132*, 18214. and *Nat. Catal.* **2020**, *3*, 516.), we focused on the kinetic barrier of deprotonation of OH_{bri} using an explicit solvent model with 1 monolayer of water. As can be seen in **Figure R9**, the deprotonation of the $\text{W-OH}_{\text{bri-Ru}}$ site showed a 0.62 eV lower barrier of deprotonation than that of $\text{Ru-OH}_{\text{bri-Ru}}$ site, which verified that introducing Brønsted acid sites could lead to faster proton transfer from the O_{bri} to the electrolyte.

Figure R9 | The kinetic barrier of the deprotonation of OH_{bri} on different catalysts with solvent. Inset: The snapshots of the initial state (IS), transition state (TS), and final state (FS) on W-doped RuO_2 .

In light of the reviewer’s comments, we have provided new Figure 5b on Page 11 in the revised manuscript and revised the following discussions:

“Since the deprotonation of OH_{bri} was regarded as the rate-limiting step in Ru-based catalysts at low overpotential, we further calculated the kinetic barrier of the deprotonation on different O_{bri} sites considering the effect of solvent (Fig. 5b). The $\text{W-OH}_{\text{bri-Ru}}$ model showed a 0.62 eV lower barrier of deprotonation compared with $\text{Ru-OH}_{\text{bri-Ru}}$, indicating a faster deprotonation process on $\text{W-OH}_{\text{bri-Ru}}$.”
(Lines 342-347, Page 12 in the revised Manuscript)

“To calculate protonation barriers of adsorbed H, we made use of the climbing image nudged elastic band (NEB) method. Explicit solvent and H_3O^+ were used to represent the electrode/electrolyte and

acid condition, respectively. A model with a monolayer of water in a hexagonal arrangement was used and the sufficient solvation of the proton by its surrounding waters, as shown in Fig. 5b. To simplify the model and explore the nature of activity enhancement, only the O_{br} site was covered by proton as the initial state during the kinetic calculation.” (Lines 426-432, Page 14 in the revised Method section)

Comment 3:

Biggest concern is the innovation of such design strategy; it lies within the broad ‘doping strategy’ and the authors have explored a plethora of doping elements.

Response:

We agree with the reviewer that our materials design is within the broad “doping strategy”, and the “doping strategy” has been proved to be very effective in improving the catalytic stability and activity. The key problem is, which element and which site or structure to dope, and how could these dopants affect the catalytic procedures. The answer to these questions will further guide the rational design of new efficient catalysts and improve the understanding of this complex OER reaction, which will be meaningful to the community.

In this manuscript, even within the broad doping strategy, we have achieved the following new advances:

- (i) From the high-level characterization, we developed and demonstrated the use of in situ electrochemical XPS in the electrocatalysis study. By such a valuable technique, we found for the first time the proton transfer during OER. Moreover, our newly designed in situ electrochemical XPS cell was versatile and flexible, which can be used to study various electrochemical reactions and can be equipped in different NAP-XPS facilities in labs and synchrotrons directly. This improves the accessibility of such advanced surface-sensitive technology to the community.
- (ii) From the scientific scope, we found the surface bridging oxygen sites are the key sites for deprotonation and found introducing Brønsted acid sites is the powerful doping strategy. The findings are much more accurate and deep than before. Moreover, this new strategy can overcome the drawbacks of some other strategies, such as facet engineering, which is

intrinsically limited to single crystal catalysts and the new thinking can be widely adopted by other electrocatalysis systems.

- (iii) From the application scope, to our best knowledge, we reported one of the most active and stable iridium-free OER catalysts in the acidic electrolytes. We also examined the stability of our catalyst under high current density, verifying the potential of the application of this new catalyst.
- (iv) From the materials chemistry scope, we reported a convenient and universal approach to synthesizing homogeneous Ru-M oxide catalysts, which can be easily scaled up.

Besides, Reviewer #1 and Reviewer #3 also praised the importance and novelty of our work. Reviewer #1 said: *“The RuOx-WOx binary catalysts show very impressive OER activity and stability. The results and general trends are important for researchers working on OER, especially in acidic environment.”* And Reviewer #2 said: *“I think this is a very good study. From a fundamental point of view, they explain how important is to tune the kinetics of proton transfer reaction in OER by adding elements with different acidity. On the other side, from an applied point of view, they report a Ru catalyst with low overvoltage and very high stability even at high current densities... and it will be of interest to the readership of this journal”.*

In sum, our work on the Brønsted acid site is not simply a “doping strategy”, but an efficient design strategy, a bottom-up materials chemistry insight to develop high-performance electrocatalysts and an improvement of high-end characterization technologies. This gave us confidence that our work will interest the broad readership of *Nature Communications*.

Response to Reviewer #3:

Comment 0:

The paper by Wen et al. combines theory and experiments to explain why introducing Brønsted acid elements in the RuO_x network enhances the proton transfer between the HOO- intermediate and oxygen bridging sites for accelerated OER. They find universality in their strategy as they observe that the OER performance on binary RuO_x catalysts correlates with the Brønsted acidity of the added metal (Cr, Nb, Ta, Ti, Mo). Moreover, they also report impressive stability of the Ru₅W₁O_x nano-catalysts. I think this is a very good study. From a fundamental point of view, they explain how important is to tune the kinetics of proton transfer reaction in OER by adding elements with different acidity. On the other side, from an applied point of view, they report a Ru catalyst with low overvoltage and very high stability even at high current densities. Nevertheless, I have several concerns regarding the electrochemical experiments in this manuscript. Once the authors address my questions, the paper will be ready for publication and it will be of interest to the readership of this journal.

Response:

We appreciate the reviewer's recommendations. To address the reviewer's concerns, we have reprocessed our electrochemical data, included new data, and added detailed discussions as suggested by the reviewer. For your convenience, the main revisions are discussed in the following point-to-point answers to the reviewer's questions and marked with yellow in the main text.

Comment 1:

I don't understand how you normalize the current against the electroactive surface area. How do you calculate the electroactive area? Is it the number of active sites on the surface, or is it the geometric area normalized by the loading of Ru? The number of active sites depends on the surface structure and the morphology of the catalyst and does not only depend on the loading of Ru. Could you explain a bit more about what BET consists of, which type of experiments do you do to determine the BET? The unities of the BET are 58.86 m² g⁻¹ on page 6, which suggest that BET is a measure of the Ru loading and not the electroactive area. Is there a change in the surface structure of the NPS when W is added to Ru that can affect/increase the electroactive area of RuWO_x?

Response:

We agree with the reviewer that the determination of the electrochemical surface area (ECSA) is critical in the performance evaluation of electrocatalysts. We will discuss the BET surface area in detail in this response and the ECSA in the response to Comment 2. We summarized the reviewer's concerns below:

1. How did you measure and calculate the BET surface area?

We measured the BET surface area using N₂ adsorption on the catalysts' powder. The BET normalized current density is calculated by:

$$j_{BET} = \frac{i}{m_{cat} \times A_{BET}}$$

where i is the OER current, m_{cat} is the mass loading of the catalyst (0.0225 mg) on the glassy carbon electrode (GCE), A_{BET} is the BET surface area of the catalyst (53.86 m² g⁻¹). So, yes, it is the geometric area normalized by the loading of the catalyst powder, according to the reviewer.

2. Why do you use BET surface area to normalize the OER current?

We agree with the reviewer that the surface area of the catalysts is not equal to the "electroactive area". But for nanomaterial catalysts, the catalyst particles are severely agglomerated. The morphology of the catalyst film on the GCE is similar to the agglomerated catalyst powder. So the BET surface area can partly reflect the electroactive area of the catalyst (the absolute value may be different, but the trend should be the same), which is an overestimated method of the electroactive area. This protocol is also recognized by some prestige groups in this field (e.g. Shao-Horn *et al. Nat. Chem.* **2017**, *9*, 457 and Xu *et al. Nat. Catal.* **2019**, *2*, 763). And for RuO₂, as early as 1977, Bruke *et al.* used BET surface area to normalize the specific OER current of RuO₂ catalyst (*J. Chem. Soc., Faraday Trans.* **1977**, *73*, 1659).

On the other hand, as mentioned by the reviewer in Comment 2 and can also be seen in **Figure R10**, when normalizing the current using the geometry area of the GCE, the current density is much larger in Ru₅W₁O_x than in the RuO₂. And this deviation can be canceled (both physically and electrochemically) when normalizing the current using BET surface area. In this circumstance, the OER activity of Ru₅W₁O_x was still higher than RuO₂, which indicated that the improvement of the activity was "intrinsic". Therefore, the BET surface area can be one of the suitable tools for our system to evaluate the intrinsic OER activity.

Figure R10 | The CV profiles. (a) Current normalized by the geometry area of the GCE. **(b)** Current normalized by oxide BET surface area.

3. Will W doping change the surface structure and affect/increase the electroactive area of RuWO_x?

Based on the above results, it can be concluded that the overall improvement of the activity in Ru₅W₁O_x was contributed by both surface area effect and intrinsic activity of active sites. The incorporation of W in RuO₂ increased both the surface area and electroactive area of the catalyst according to our results.

The evaluation of ECSA will be discussed in the response to Comment 2.

In light of the reviewer’s comments, we have revised the Fig. 3a on Page 7 in the revised manuscript and revised the following discussions:

“The specific activity of Ru₅W₁O_x was obtained by normalizing the OER current using either the catalyst’s BET surface area or the mercury underpotential deposition (Hg-UPD) determined electrochemical active surface area (ECSA) (Supplementary Fig. 10). Both values surpassed the pristine RuO₂ by *ca.* 2 times at 1.50 V vs. RHE (Supplementary Fig. 11).” **(Lines 170-174, Page 6 in the revised Manuscript)**

“The above results verified that the incorporation of W-O_{br}-Ru Brønsted acid sites improved the OER activity of RuO₂ both apparently (by the increase of electroactive surface area) and intrinsically (by the increase of reactivity of per active site), indicating a lower barrier and a different OER mechanism for Ru₅W₁O_x.” **(Lines 176-180, Page 6 in the revised Manuscript)**

Comment 2:

The pseudocapacitive region of $\text{Ru}_5\text{W}_1\text{O}_x$ is considerably larger than the RuO_x pseudocapacitive region, in figure 15 of the supporting information. The pseudocapacitive region does not provide an exact measure of the increase in electroactive surface area, but it could suggest that there is a change in the active area when W is introduced in the RuO_x network. It is hardly difficult to calculate the electroactive surface active area of IrO_x and RuO_x based-catalysts (maybe you can check this reference <https://doi.org/10.1021/acscatal.7b03787>). Could you discuss the differences between ECSA and specific activity (normalized by loading) and the limitations on the determination of the electroactive area in these metal oxides? I agree with the author's history and the importance of the Bronsted acidity of the added elements, and it is evident that W makes RuO_x incredibly stable. However, as you work with non-well-defined nanoparticles, you should briefly mention that changes in the surface structure can also influence the geometric activity.

Response:

We have summarized the reviewer's concerns below:

1. Determining the electroactive surface area using the Hg-UPD method.

Figure R11 | Determining the ECSA using mercury underpotential deposition. The cyclic voltammetry of (a) $\text{Ru}_5\text{W}_1\text{O}_x$ and (b) RuO_2 at 50 mV s^{-1} scan rate. The experiment was conducted in 0.1 M HClO_4 containing $1 \text{ mM Hg(NO}_3)_2$. The current difference of the cathodic scans between the Hg-containing solution and blank background was integrated to calculate the amount of Hg_{upd} . A coulombic charge of $138.6 \mu\text{C cm}^{-2}$ was used as a factor to obtain the Hg_{upd} -ECSA values.

As the reviewer suggested, we carried out the mercury underpotential deposition (Hg-UPD) experiments to determine the ECSA of different catalysts (**Figure R11** and **Table R1**). According to the results, the $\text{Ru}_5\text{W}_1\text{O}_x$ showed a higher ECSA value ($33.78 \text{ m}^2 \text{ g}^{-1}$) than the RuO_2 ($8.03 \text{ m}^2 \text{ g}^{-1}$) and the specific OER activity showed a similar trend with the BET normalization (**Figure R12**). Besides, we observed two different UPD potentials on the $\text{Ru}_5\text{W}_1\text{O}_x$, indicating the surface structure of RuO_2 was changed after W incorporation by generating different surface sites. Therefore, we concluded that the improvement of the overall activity of the Ru-W catalyst was contributed by both the increase of the electroactive surface area and the improvement of the intrinsic activity of new active sites, which is the importance of the Bronsted acidity of the added elements.

Figure R12 | Evaluation of specific OER activity. (a) Normalized by ECSA (determined by Hg-UPD method). **(b)** Normalized by BET surface area of the catalyst powder.

2. What were the limitations on the determination of the ECSA in Ru/Ir metal oxides?

The traditional method to evaluate the electroactive area of nano-catalysts is using the electrochemical double-layer capacitance (C_{dl}). And the typical method to measure the C_{dl} is scanning CVs in non-Faradic regions at different scan rates. We also conducted the C_{dl} measurements using cyclic voltammetry accordingly (**Figure R13**). The general trend of the C_{dl} -derived ECSA ($33.4 \text{ m}^2 \text{ g}^{-1}$ to $76.1 \text{ m}^2 \text{ g}^{-1}$ after W doping) was the same as the BET ($9.85 \text{ m}^2 \text{ g}^{-1}$ to $53.86 \text{ m}^2 \text{ g}^{-1}$ after W doping) and Hg-UPD ($8.03 \text{ m}^2 \text{ g}^{-1}$ to $33.78 \text{ m}^2 \text{ g}^{-1}$ after W doping) methods, but the absolute value was higher. We attribute this to the pseudocapacitive current contributions on the Ru/Ir-based oxides, which will lead to an overestimation of the ECSA.

Based on the above results, we can summarize the limitations of different methods to determine the specific activity of OER catalysts:

- (i) The BET surface area (normalized by loading) is a universal method for the powder catalysts. But since not all the areas could contribute to the electroactivity, it is an overestimated method for the actual ECSA. Besides, the measurement of BET surface area is based on the physical adsorption of N_2 molecules, which is different from the chemical adsorption of reactants at the catalyst/electrolyte interface.
- (ii) The electrochemical double-layer capacitance method is convenient and suitable for some materials. But this method does not take into account other possible contributions to the measured capacitance, e.g. the pseudocapacitance in Ru/Ir oxides. Besides, the double-layer capacitance measurements assume that the metal oxide catalysts are equally conductive, which is another potential source of error in these measurements.
- (iii) The underpotential deposition methods are relatively accurate for catalyst/electrolyte systems. But the selection of UPD elements varied from material to material and lack of universal procedures for different catalysts. This method can be more suitable and reliable than electrochemical double-layer capacitance method when the pseudocapacitance is large.

Therefore, we provide the results of all three methods in the revised manuscript for a comprehensive comparison (**Table R1**). We revised Fig. 2b (also see **Figure R14**) in the revised manuscript, and add a new axis demonstrating the specific OER current density normalized by Hg-UPD surface area. We also included a new Supplementary Note 1 to discuss the determination of specific activity in the revised Supplementary Information.

Figure R13 | Determination of C_{dl} by cyclic voltammetry. (a and b) $Ru_5W_1O_x$. (c and d) RuO_2 . (e

and f) IrO₂. Cyclic voltammetry scanned between 0.20 to 0.30 V vs. MSE. The cathodic and anodic charging currents measured at 0.25 V vs. MSE were plotted as a function of scan rate. A general specific capacitance (C_s) of 35 $\mu\text{F cm}^{-2}$ was used to calculate ECSA.

Figure R14 | Summary of some major OER performance metrics of Ru₅W₁O_x and RuO₂. The TOF and specific OER activity (j_{spec}) (normalized by BET surface area and Hg-UPD surface area respectively) were calculated at 1.50 V vs. RHE. The apparent activation energy (E_a) was calculated by the OER current of 1.50 V vs. RHE at different temperatures. The mass-specific activity was calculated at $\eta = 300$ mV based on total metal loading.

Table R1 | Summary of the surface area obtained by different methods. We converted all the values to the same unit for a direct comparison.

Catalyst	S_{BET} (m ² g ⁻¹)	¹ $S_{\text{Hg-UPD}}$ (m ² g ⁻¹)	² S_{CdI} (m ² g ⁻¹)
Ru ₅ W ₁ O _x	53.86	33.78	76.1
RuO ₂	9.85	8.03	33.4
IrO ₂	11.98	6.48	24.2

¹The surface area factor is 138.6 $\mu\text{C cm}^{-2}$.

²The surface area factor is 35 $\mu\text{F cm}^{-2}$.

In light of the reviewer’s comments, we have revised the Fig. 3a on Page 7 in the revised manuscript and revised the following discussions:

“The specific activity of Ru₅W₁O_x was obtained by normalizing the OER current using either the catalyst’s BET surface area or the mercury underpotential deposition (Hg-UPD) determined

electrochemical active surface area (ECSA) (Supplementary Fig. 10). Both values surpassed the pristine RuO₂ by *ca.* 2 times at 1.50 V vs. RHE (Supplementary Fig. 11).” (Lines 170-174, Page 6 in the revised Manuscript)

“The above results verified that the incorporation of W-O_{bri}-Ru Brønsted acid sites improved the OER activity of RuO₂ both apparently (by the increase of electroactive surface area) and intrinsically (by the increase of reactivity of per active site), indicating a lower barrier and a different OER mechanism for Ru₅W₁O_x.” (Lines 176-180, Page 6 in the revised Manuscript)

We have also included a new Supplementary Note 1 to discuss the determination of the electroactive area in detail.

Comment 3:

I can see that the authors have calculated the mass activity normalizing the current density against the loading of Ru in figure 10Sa. However, Figure 14S shows that WO_x has the OER onset at E < 1.500 V vs RHE. could W surrounded by Ru contribute to the OER? Why do not you normalize the current against the loading of both Ru+W? I guess that W contributes very little to OER, but it is also a rare element and expensive, thus I find it highly necessary to show what the mass activity is concerning both elements and not only Ru. I would suggest doing an additional figure, similar to Figure 14S, with the mass-specific activity (A g⁻¹ Ru+W) and discussing the results. The same applies to the other elements (Cr, Mo, Nb). Apart from figure 6a (manuscript), I would show a figure in which I would represent the mass-specific activity considering both elements (A g⁻¹ (Ru+ M), M=Cr, Nb...). Figure 6b (manuscript).

Response:

We summarized the reviewer’s concerns as below:

1. Will the W surrounded by Ru contribute to the OER?

Figure R15 | The OER polarization curves of different catalysts.

To address the reviewer’s concern about the OER activity of W sites, we plotted the LSV curve of WO_3 with the Ru-based catalysts (**Figure R15**). The WO_3 showed almost no OER reactivity. However, the WO_3 did not show the OER activity, meaning that the W atoms in the RuO_2 matrix do not contribute the OER activity.

2. Normalize the current against the loading of Ru+W

Figure R16 | The OER performances of different catalysts. (a) The mass-specific activity of different catalysts at $\eta = 300$ mV. **(b)** The TOF values of different catalysts.

We agree that the reviewer’s suggestions on the identification of the active site and mass-specific activity calculation are reasonable. Therefore, we re-calculated the mass-specific activity and TOF of our catalysts based on the total metal atom loading (Ru+W) (**Figure R16**). Based on the new calculations, the absolute values of mass-specific activity decreased, but the general trends were

retained the same as before.

3. Could you calculate the mass-specific activity of Ru-M catalysts using the total metal loading?

As suggested, we calculated the mass-specific activity and TOF values of Ru-M catalysts in Figure 6 based on the total metal loading (Ru+M) (Figure R17). The activity trend was also maintained the same as before in the new figures.

Figure R17 | The OER performance of Ru-M catalysts. (a) The mass-specific activity (calculated by total metal loading). **(b)** The relationship between TOF (calculated by total metal loading) and E_{ads} of H⁺ on bridging oxygen site.

In light of the reviewer’s comments, we have revised the Supplementary Fig. 8 and 14 in revised Supplementary Information, added new Fig. 6b on Page 7 in the revised manuscript, and revised the following discussions:

“The mass-specific activity was improved by 8-fold (750 A g_{Ru}⁻¹ of Ru₅W₁O_x vs. 87 A g_{Ru}⁻¹ of RuO₂, estimated by total Ru loading mass). When calculated by the total metal loading (Ru+W), the mass-specific activity of Ru₅W₁O_x was 547 A g_{metal}⁻¹, 6 times higher than the RuO₂ (Supplementary Fig. 8).” (Lines 165-168, Page 6 in the revised Manuscript)

Comment 4:

Regarding the electrochemical or LSV experiments, in figure 3a (manuscript). How many cycles or scans have you taken? How do the LSV curves change after consecutive scans? The LSV of RuO_x looks

like the RuO_x is getting dissolved when you achieve a high current of $40\text{--}50\text{ mA cm}^{-2}$ because it weirdly crosses the other LSV. Could you indicate the stability of the cycles and the cycle number you represent in the figure?

Response:

Before the LSV measurements, we conducted 10 consecutive CV cycles to clean and stabilize the surface. As shown in **Figure R18**, the $\text{Ru}_5\text{W}_1\text{O}_x$ kept stable within 10 cycles, and no obvious decrease was observed. While the RuO_2 degraded along with the cycle number increase. The above results indicated that the stability was much improved in the W-doped catalyst.

Figure R18 | Ten CV cycles of different catalysts before the LSV test. (a) $\text{Ru}_5\text{W}_1\text{O}_x$. (b) RuO_2 . Scan rate: 50 mV s^{-1} .

In addition, we carried out intense CV cycles on the $\text{Ru}_5\text{W}_1\text{O}_x$ to further investigate the stability of the catalyst. After performing 20,000 cycles between 0.95 to 1.55 V vs. RHE, the catalyst only showed a slight potential degradation in the OER activity (**Figure R19**).

Figure R19 | The LSV curves of $\text{Ru}_5\text{W}_1\text{O}_x$ before and after 20,000 CV cycles between 0.95 to 1.55 V vs. RHE.

In light of the reviewer’s comments, we have added new Supplementary Fig. 12 in revised Supplementary Information, and revised the following discussions:

“We also performed the same test on the commercial IrO_2 catalyst (~ 5 nm, BET surface area $11.98 \text{ m}^2 \text{ g}^{-1}$, Supplementary Fig. 15), which was highly stable as expected. And the $\text{Ru}_5\text{W}_1\text{O}_x$ has shown comparable stability to state-of-the-art IrO_2 catalysts. In the intense cycle test, the $\text{Ru}_5\text{W}_1\text{O}_x$ could also stay active even after 20,000 CV cycles (Supplementary Fig. 16). While the RuO_2 showed poor stability in both chronopotentiometry and cycle tests (Supplementary Fig. 17).” **(Lines 187-193, Page 6 in the revised Manuscript)**

“To evaluate the OER activity of different catalysts, the WEs were first performed 10 CV cycles between 0.95 to 1.50 V vs. RHE (before iR-correction) at 50 mV s^{-1} scan rate to clean and stabilize the surface, then followed with a LSV scan from 0.95 to 1.65 V at 5 mV s^{-1} scan rate.” **(Lines 485-488, Page 15 in the revised Method section)**

Comment 5:

Why have you selected H_2SO_4 as an electrolyte rather than HClO_4 ? Can the adsorption of sulfate affect the catalyst performance because of sulfate adsorption?

Response:

We selected H₂SO₄ as electrolyte for no specific reason. H₂SO₄ is a stable and widely used electrolyte in the evaluation of OER catalysts in acidic environments (e.g. *Science* **2016**, 353, 1011; *Nature* **2020**, 587, 408; *Nat. Catal.* **2018**, 1, 841).

As suggested, we carried out the electrochemical measurements in 1 M HClO₄ (similar pH with 0.5 M H₂SO₄) and no significant difference was observed in our experiments (**Figure R20**). The OER activity and the surface electrochemical behavior of our catalyst are similar in both electrolytes. We also noticed some reports on the electrolyte effect in the OER performance of Ir-based nanoparticles (*ChemPhysChem* **2019**, 20, 2956 and *Nat. Commun.* **2021**, 12, 6007). In these reports, the Ir-based catalysts showed higher OER activity in the HClO₄ than in the H₂SO₄. The adsorption of sulfate may interfere with the oxidation of Ir, leading to lower OER activity. However, in our case, we did not observe a similar electrolyte effect.

On the other hand, in our new experiment, we noticed that the anions do affect the stability measurement, as will be discussed in Comment 6.

Figure R20 | The electrochemistry of Ru₅W₁O_x in different electrolytes. (a) LSV curves. Scan rate: 5 mV s⁻¹. (b) CV curves. Scan rate: 200 mV s⁻¹.

In light of the reviewer’s comments, we have added new Supplementary Fig. 15 in the revised Supplementary Information, and revised the following discussions:

“We also checked the electrochemical behavior of Ru₅W₁O_x in 1 M HClO₄ (same pH as 0.5 M H₂SO₄). No obvious electrolyte effect could be observed, which indicates that the adsorption of sulfate

will not interfere with the surface chemistry of Ru, different from the Ir-based catalysts (Supplementary Fig. 22).” (Lines 224-228, Page 8 in the revised Manuscript)

Comment 6:

The stability of RuWO_x is extraordinary. Figure S13 shows the stability experiment at 100 mA cm⁻² for 24 hours. Why not show the whole curve until the dissolution of the RuWO_x catalyst? In addition, how can the corrosion of glassy carbon affect stability at very high current values and overpotential?

Response:

As suggested by the reviewer, we showed the full v-t curve of the stability test at 100 mA cm⁻² (Figure R21). The stability showed a drastic drop after a 40-h test. And as the reviewer mentioned, the carbon corrosion was severe at a very high current density, which led to the instability of the carbon substrate and a misleading of the catalyst stability. This phenomenon is not sensitive at low current density but will be more prominent at high current density.

Figure R21 | The chronopotentiometry of Ru₅W₁O_x at 100 mA cm⁻². Using carbon paper as the substrate, without iR compensation. The drastic activity drop was mainly due to the carbon corrosion at high current density in H₂SO₄.

Meanwhile, we carried out a new high current density stability test by changing the electrolyte to 1 M HClO₄. We found that the stability was much improved. This result verified the stability of our W-doped catalyst. We attribute the improvement of stability to the intercalation of anions into carbon

materials. The intercalated anion transport into the bulk of the carbon substrate and is protected against a reaction with the aqueous electrolyte (*Electrochim. Acta* **1981**, 26, 799; *J. Electrochem. Soc.* **2000**, 147, 2636). Since the ClO_4^- ions intercalate easier than the HSO_4^- ions, the carbon substrate will be more stable in HClO_4 .

In light of the reviewer's comments, we have revised Supplementary Fig. 13 in revised Supplementary Information, and revised the following discussions:

“At higher electrolysis current densities (100 mA cm^{-2}), the stability of $\text{Ru}_5\text{W}_1\text{O}_x$ was also maintained within a 100-h test (Supplementary Fig. 20).” **(Lines 197-198, Page 7 in the revised Manuscript)**

Minor comment:

Page 9 and 10. I think you may refer to Figures 4 e and 4 d when explaining the NAP-XPS experiment? I would suggest changing the order of figures so that 4 d is RuO_2 and 4 e is $\text{Ru}_5\text{O}_1\text{O}_x$ because the RuO_2 is explained at the beginning of this section.

Response:

We feel sorry for the mistake. As suggested by the reviewers, we have fixed the notation and exchanged the order of Fig. 3d and 3e (the original Fig. 4d and 4e). Meanwhile as suggested by Reviewer #1, we removed Fig. 1 and re-organized Fig. 5 and Fig. 6 to form a coherent DFT part.

REVIEWER COMMENTS

Reviewer #1 (Remarks to the Author):

The authors have adequately addressed all my comments. I recommend acceptance.

Reviewer #2 (Remarks to the Author):

This is a well-prepared response which addresses many concerns from all reviewers. The innovation is better clarified and convince. Some technical questions which needs serious consideration before acceptance.

1. The authors need to give more information about how the H⁺ energy was calculated in Figure R2(a), especially how the positive charge was modelled. The authors also need to confirm that the adsorbed H⁺ possess one positive charge.

2. Water desorption barrier is very high; even for the W-doped RuO₂ model it's as high as 3.76 eV. Considering that normally only reactions showing barrier value less than 0.75 eV is considered achievable under room temperature, the authors need to give solid reason how they conclude such barrier for deprotonation of OH_{bri} is surmountable at room temperature.

Reviewer #3 (Remarks to the Author):

I think that all my comments have been very-well addressed and discussed. The authors have invested time and a huge effort in improving the discussion of the results, adding new interesting data. I think the paper is of high quality, it will be very well received by the readers of the journal and should be published.

Actions (regular font) in Response to Reviewer Comments (*italics*)

Reviewer #1 (Remarks to the Author):

The authors have adequately addressed all my comments. I recommend acceptance.

Response:

We thank the reviewer's recommendation.

Reviewer #2 (Remarks to the Author):

Comment 0:

This is a well-prepared response which addresses many concerns from all reviewers. The innovation is better clarified and convince. Some technical questions which needs serious consideration before acceptance.

Response:

We thank the reviewer's advice. We have revised the manuscript and carried out new calculations to address the reviewer's concerns.

Comment 1:

The authors need to give more information about how the H^+ energy was calculated in Figure R2(a), especially how the positive charge was modelled. The authors also need to confirm that the adsorbed H^+ possess one positive charge.

Response:

We thank the referee for the queries. We calculated the adsorption energy based on the following equations:

$$E_{ads} = E(OH_{bri}) - E(O_{bri}) - E(H^+ + e^-),$$

in which the energy of proton and electron, based on the CHE model (*Energy Environ. Sci.* **2010**, *3*, 1311, *J. Phys. Chem. B* **2004**, *108*, 17886), can be simplified by:

Therefore, the hydrogen atom is not charged in the calculations.

To clarify the above calculations, we revised the y-axis label of Fig. 5a (also see **Figure R1**).

In the new AIMD calculation to evaluate the energy barrier, we model the H^+ with one positive charge and the counter ion, which is different from the CHE model above. The details are in the following response to Comment 2.

Figure R1 | The H atom adsorption energy on different surface O_{bri} sites. Inset: Schematic of different O_{bri} sites on the W-doped RuO_2 .

In light of the reviewer’s comments, we revised the following discussions, the corresponding figure captions and axis labels:

“We then examined the adsorption energy (E_{ads}) of H (by assuming the energy of $H^+ + e^-$ as the energy of $1/2 H_2$ molecule) on different O_{bri} sites.” (Lines 335-336, Page 12 in the revised Manuscript)

We also changed the word “proton” to “hydrogen atom” or “H” in the DFT part.

Comment 2:

Water desorption barrier is very high; even for the W-doped RuO_2 model it’s as high as 3.76 eV. Considering that normally only reactions showing barrier value less than 0.75 eV is considered achievable under room temperature, the authors need to give solid reason how they conclude such barrier for deprotonation of OH_{bri} is surmountable at room temperature.

Response:

In the previous calculations, to estimate the hydrogen desorption barrier, we simplified the model (with only one layer of water molecules) for comparison and obtained 3.76 eV, which was high. Now we use *ab initio* molecular dynamics (AIMD) calculations at 298 K to optimize the interactions between the multiple water layers and oxide surface (similar to the structures used in *Nature* 2020,

587, 408 and *ACS Catal.* **2015**, *5*, 2317), perform new calculations to evaluate the barriers, and obtain reasonable values.

As shown in **Figure R2**, there is a vacuum in the previous model with only one layer of water molecules. Thus we have to constrain most of the water molecules to compare the desorption barrier quickly. It leads to high energy barrier values for both the RuO₂ model and the W-doped RuO₂ model. Compared to the multiple water layers (**Figure R2b and R2b'**), one of the water molecules could reach the hydrogen atom on the surface of the substrate with 1.89 Å. In comparison, the water molecule is far from the hydrogen atom on the surface of the substrate for the one-layer water model, which leads to a high energy barrier.

Figure R2 | Configurations of initial states of the proton transfer on W-doped RuO₂ surface. (a) the previous monolayer water model and (b) the 3D explicit water model optimized by AIMD.

Then, we carry out the kinetics barrier calculations of deprotonation based on this model (**Figure R3**). The details of the model are presented in the new Supplementary Note 5 in the revised Supplementary Information.

Figure R3 | The explicit water models of water/(W-doped)RuO₂ interface. (a) Example of cell with explicit water used to compute surface phase diagram. The optimized configurations of the initial state (IS), the transition state (TS), and the final state (FS) of the proton transfer from the surface of (b-d)

RuO₂ and (e-g) W-doped RuO₂. The transferred proton is highlighted. The dashed yellow lines between water molecules indicate hydrogen bond interactions. Ru atoms are dark green, W as blue, O as red, and H as white.

Figure R4 | The kinetic barrier of the deprotonation of OH_{bri} on different catalysts with solvent. Insets: The snapshots of the initial state (IS), transition state (TS), and final state (FS) on W-doped RuO₂.

As shown in **Figure R4**, in the new models, the deprotonation barrier on the O_{bri} site of W-doped RuO₂ is 0.11 eV, which is lower than that of RuO₂ (0.23 eV). Surface charge distribution calculation shows that H atoms on the RuW surface (+0.144 |e|) present higher positive charges than the H atoms on the RuO₂ surface (+0.116 |e|), which can support the conclusion that H atoms adsorbed on W-O_{bri}-Ru are more likely to be deprotonated than those adsorbed on Ru-O_{bri}-Ru.

In addition, the FS state shows lower energy than IS state. The reason is that H₃O⁺ is an energy-favorable state. There is a balance between H₃O⁺ and H₅O₂⁺, where H₅O₂⁺ shows higher energy than H₃O⁺. (See *Science* **2022**, 377, 315.)

In light of the reviewer’s comments, we revised Fig. 5b and added new Supplementary Note 5, Supplementary Fig. 43 and 44 in the revised manuscript and Supplementary Information. The corresponding discussion and experimental details are also presented in the revised manuscript.

“..., we further calculated the kinetic barrier of the deprotonation on different O_{bri} sites considering the effect of solvent (Supplementary Note 5, Supplementary Fig. 43 and 44). The W-OH_{bri}-Ru model showed a lower barrier of deprotonation compared with Ru-OH_{bri}-Ru, indicating a faster deprotonation process on W-OH_{bri}-Ru (Fig. 5b)” **(Lines 344-348, Page 12 in the revised Manuscript)**

“To simulate the interaction at the water/(W-doped)RuO₂ interface, we used 18 explicit water molecules (6 layers) on a 2 × 1 RuO₂ surface slab (3 layers) with an area of 6.28 × 6.42 Å². The

simulation box is 28 Å along the z-axis. The initial structure of the water box is based on the density of the solvent^{42,43} (as shown in Supplementary Fig. 43). To equilibrate the waters interacting with the interface, we carried out 850 steps (time step is 1 fs) of *ab initio* molecular dynamics (AIMD) simulation at 298 K⁴⁴. The temperature and potential energies during the AIMD simulation are shown in Supplementary Fig. 44. To calculate deprotonation barriers of adsorbed H, we made use of the climbing image nudged elastic band (CI-NEB) method⁴⁵ based on the established models.” (Lines 427-436, Page 14 in the revised Manuscript)

Reviewer #3 (Remarks to the Author):

I think that all my comments have been very-well addressed and discussed. The authors have invested time and a huge effort in improving the discussion of the results, adding new interesting data. I think the paper is of high quality, it will be very well received by the readers of the journal and should be published.

Response:

We thank the reviewer's recommendation.

REVIEWERS' COMMENTS

Reviewer #2 (Remarks to the Author):

The authors have performed additional calculations to confirm the energy barrier, which is now reasonable. Given this critical concern have been addressed, I would recommend acceptance.

ROUND #3

REVIEWERS' COMMENTS

Reviewer #2 (Remarks to the Author):

The authors have performed additional calculations to confirm the energy barrier, which is now reasonable. Given this critical concern have been addressed, I would recommend acceptance.

Response:

We thank the reviewer's recommendation.